# STATISTICAL GUARANTEES FOR CONSENSUS CLUSTERING

**Zhixin Zhou**[1], **Gautam Dudeja**[2], **Arash A. Amini**[2]

[1]City University of Hong Kong,    [2]University of California, Los Angeles

zhixin0825@gmail.com,   gdudeja@g.ucla.edu,   aaamini@ucla.edu.

## ABSTRACT

Consider the problem of clustering $n$ objects. One can apply multiple algorithms to produce $N$ potentially different clusterings of the same objects, that is, partitions of the $n$ objects into $K$ groups. Even a single randomized algorithm can output different clusterings. This often happens when one samples from the posterior of a Bayesian model, or runs multiple MCMC chains from random initializations. A natural task is then to form a consensus among these different clusterings. The challenge in an unsupervised setting is that the optimal matching between clusters of different inputs is unknown. We model this problem as finding a barycenter (also known as Fréchet mean) relative to the misclassification rate. We show that by lifting the problem to the space of association matrices, one can derive aggregation algorithms that circumvent the knowledge of the optimal matchings. We analyze the statistical performance of aggregation algorithms under a stochastic label perturbation model, and show that a $K$-means type algorithm followed by a local refinement step can achieve near optimal performance, with a rate that decays exponentially fast in $N$. Numerical experiments show the effectiveness of the proposed methods.

## 1 INTRODUCTION

Clustering is a fundamental task in machine learning and data analysis. Given data on each of the $n$ objects in a set, there are numerous algorithms to produce a clustering of these $n$ objects, which is formally a partitioning of $\{1, \ldots, n\}$ into $K$ disjoint sets. A natural problem that arises in practice is how to form a consensus among these clusterings. This is especially important if the different clusterings are produced by a single randomized algorithm. This situation often arises in Bayesian modeling, where the posterior naturally encodes the variability of the clustering problem. Finding a consensus clustering then corresponds to finding the center of the posterior, from which we can also obtain estimates of the variability of the posterior.

A clustering of $n$ objects can be viewed as a label vector in $[K]^n$ where $[K] = \{1, \ldots, K\}$. We assume that we are given $N$ label vectors $z_j \in [K_j]^n$ for $j = 1, \ldots, N$, with potentially different number of clusters each. Let $K = \max_j K_j$ and note that we can view all $z_j$ as vectors in $[K]^n$. The task is to obtain a consensus $K$-clustering, that is, a label vector $z \in [K]^n$ which is close to all $z_1, \ldots, z_N$ at the same time. We also refer to this task as the *label aggregation* problem.

In the context of clustering, there is no meaning to the label of each cluster, that is, the label aggregation problem is *unsupervised*, in the sense that there is no natural correspondence between labels of different clusterings. This is in contrast to label aggregation in classification in which the labels have a common meaning among different input classifications. We refer to the latter task as *supervised* label aggregation.

In the unsupervised setting, forming a consensus label is a nontrivial task due the label-switching problem. Consider for example, the case $n = 5$ and the two label vectors $z_1 = (1, 1, 1, 2, 2)$ and $z_2 = (2, 2, 2, 1, 1)$. These two vectors are different in all 5 positions but they define the same clusterings of the objects. In this case, the consensus label $\widehat{z}$ can be taken to be either $z_1$ or $z_2$. More generally, for every $z_j$, there could be a permutation $\pi_j$ on $[K]$, such that the permuted vectors $\pi_j \circ z_j := (\pi_j(z_{ji}))_{i=1}^n$, are closer to each other than the original $z_j$s.

To formalize the above idea, we recall the definition of the misclassification rate between two label vectors, $z, y \in [K]^n$:

$$\text{Mis}(z, y) = \min_{\pi} \frac{1}{n} \sum_{i=1}^{n} 1\{z_i \neq \pi(y_i)\} \tag{1}$$

where the minimum is taken over all the permutations $\pi : [K] \to [K]$. $\text{Mis}(\cdot, \cdot)$ is a proper metric on the space of $K$-clusterings of $n$ objects. It is also a metric on $[K]^n$ if we identify vectors that are obtained from each other by label-switching. We can now define the consensus label as the barycenter of $z_1, \ldots, z_N$ in $\text{Mis}(\cdot, \cdot)$ metric, that is,

$$\widehat{z} \in \underset{z \in [K]^n}{\arg\min} \sum_{j=1}^{N} w_j \, \text{Mis}(z, z_j) \tag{2}$$

where $w_j \geq 0$ are a given set of weights. We often assume uniform weights: $w_j = 1$ for all $j$. The barycenter $\widehat{z}$ is also known as the Frechét mean. Solving (2) is complicated by the presence of the permutation in the definition of $\text{Mis}$ function. More explicitly, we need to solve

$$\widehat{z} \in \underset{z \in [K]^n}{\arg\min} \, \min_{\pi_1, \ldots, \pi_N} \sum_{j=1}^{N} \sum_{i=1}^{n} w_j 1\{z_i \neq \pi_j(z_{ji})\} \tag{3}$$

showing that in addition to $z$, we have to optimize over $N$ permutations $\pi_j, j = 1, \ldots, N$. In this paper, we provide alternative solutions that avoid optimizing over these permutations.

**Our contributions**   The unsupervised version of the label aggregation problem is the realistic and practical one when dealing with aggregating labels from Bayesian clustering algorithms, since the posterior has $K!$ modes corresponding to all possible label permutations, and the output will be near an arbitrary mode in each run of the algorithm. The main contributions of this paper to unsupervised aggregation are the following:

1. We show that by lifting the barycenter problem to the space of association matrices, one can derive algorithms that avoid optimizing over the unknown permutations (Section 2.1). In particular, we propose both a basic and a spectral $K$-means type aggregation algorithm.

2. We propose a random perturbation model (RPM) under which we can study the theoretical performance of both supervised and unsupervised aggregation algorithms. We prove the *statistical consistency* of the basic aggregation algorithm under RPM (Section 2.2).

3. Under RPM, the supervised setting corresponds to an oracle that knows the true matching permutations. By studying this oracle, we derive the optimal statistical misclassification rate for supervised aggregation (Section 3.1).

4. We propose an efficient *local refinement* step on the output of any consistent aggregation algorithm in the unsupervised setting, and show that the updated labels achieve nearly the same misclassification rate as the above oracle (Section 3.2).

Our theoretical analysis illustrates how different parameters affect the difficulty of the label aggregation problem. In Section 4, we provide numerical experiments comparing the performance of the proposed algorithms against each other and existing methods.

**Related work**   In the supervised setting, the problem of label aggregation is to combine multiple annotated dataset. The label inferred for each item from those produced by multiple annotators acts as the ground truth for the classification task. Various probabilistic models have been proposed for aggregating annotations, with parameters to account for the expertise of the annotators and the noise in the labeling process (47; 37). The unsupervised setting is more challenging as there is no meaning to the cluster labels (the label-switching issue) and the clusterings can have potentially different number of clusters. The idea of passing to association matrices to get around the label-switching issue, has been leveraged in several existing approaches (13; 24; 43; 13; 21; 29), although the connection we make to the lifted barycenter problem and the resulting spectral methods is new to the best of our knowledge. In (24; 43), the authors employ an Expectation-Maximization strategy to obtain a nonnegative matrix factorization of the combined association matrix. The authors of (41) provide

several approaches, using the hypergraph representation of the clusterings, that have shown promising results in the context of image segmentation (22). A set of fuzzy consensus algorithms is proposed in (40) that generate soft consensus partitions by combining a collection of fuzzy clusterings.

What we referred to as unsupervised label aggregation problem has appeared under many different names in the literature, including but not limited to, cluster ensembles (41), clustering/cluster ensemble problem (44; 10), ensemble clustering (1), clustering aggregation (17), combining clusterings (42; 14; 27), consensus clustering (48; 18) and the median partition problem (12; 46; 18). As can be surmised from the variety of names, there is a copious literature on the subject, spanning over multiple fields, with many ideas rediscovered time after time. We refer to the excellent surveys (48; 44; 18) for a more exhaustive list of references and historical discussions. There is also a parallel line of work in the Bayesian clustering literature on aggregating the posterior clusterings (31; 7; 9; 25; 15; 45; 8).

The barycentric view to aggregation that we take in (2) has appeared in many previous work, but often with a different distance in place of Mis, including but not limited to the symmetric difference distance (SDD), a.k.a. the Mirkin metric (up to a constant), in the median partition problem (39; 23; 12; 17; 18), the Binder loss and variational information (VI) in (45; 15), the normalized mutual information (NMI) in the pioneering work of (41), the adjusted Rand index (15) and the category utility function (42; 35). After introducing our methods, in Section 2.3, we give a more detailed comparison with the literature. We choose Mis as the distance in the present work for a better comparison with the oracle problem in Section 3.1. We also show in Appendix B.1 that consistency in Mis implies the consistency in other distances.

Despite the voluminous literature on the subject, statistical analysis of the methods under a statistical model for the input clusterings has not been undertaken before. This is the gap that we fill in this paper, by providing the first consistency and optimality results under a statistical model (the RPM) for a method of clustering aggregation that we propose. To the best of our knowledge, the question of consistency, let alone optimality, has not been considered for any method of aggregation before. We also shed more light on the relation between the barycenteric approach and those based on association matrices (Section 2.3), and how convex relaxation leading to a spectral method can be used to approximate the median partition. To illustrate the importance of statistical analysis, we also show that a simple common approach to the median partition problem, known as the BestOfK (18), is in general inconsistent under RPM, despite being shown to be a 2-factor approximation of the median partition problem (12). This further highlights the key insights that statistical analysis under a model can provide which is not possible to obtain by CS-type theory on approximation algorithms.

## 2 Lifted Aggregation Algorithms

We start by introducing some notation. Let $\mathcal{E}_K = \{e_k\}_{k=1}^K$ be the set of standard basis vectors of $\mathbb{R}^K$. The elements of $\mathcal{E}_K$ can be considered one-hot encodings of the labels from $[K]$. From now on, instead of encoding labels as element of $[K]$, we encode them as element of $\mathcal{E}_k$. We can then view labelings of $n$ objects as elements of the following set

$$\mathcal{E}_K^n = \{z = (z_1, \ldots, z_n) : z_i \in \mathcal{E}_K \ \forall i \in [n]\}. \tag{4}$$

Each $z_i$ is viewed as a $K \times 1$ vector and each element of $\mathcal{E}_K^n$ as $K \times n$ matrices, which we refer to as label matrices. For $Z \in \mathcal{E}_K$, permuting the cluster labels is equivalent to pre-multiplication by a $K \times K$ permutation matrix $P$, that is, $PZ$.

The label aggregation problem can be restated as follows: Given label matrices $Z_1, \ldots, Z_N \in \mathcal{E}_K^n$, find a consensus label matrix $Z \in \mathcal{E}_K^n$, by solving the barycenter problem:

$$\widehat{Z} \in \underset{Z \in \mathcal{E}_K^n}{\operatorname{argmin}} \ \min_{P_1, \ldots, P_N} \sum_{j=1}^N w_j \|Z - P_j Z_j\|_F^2 \tag{5}$$

where $P_1, \ldots, P_N$ are $K \times K$ permutation matrices and $\|X\|_F := \left( \sum_{i,j} X_{ij}^2 \right)^{1/2}$ is the Frobenius norm of matrix $X$. One can verify that problem (5) is equivalent to (3). The following result shows that if we know the optimal permutations $P_j$s, we can easily find the barycenter $\widehat{Z}$:

**Proposition 1.** *Let $\{\widehat{P}_1, \ldots, \widehat{P}_N\}$ be an optimal set of permutation matrices in (5). Then, the optimal solution $\widehat{Z}$ of (5) is the columnwise "argmax" of $\sum_j w_j \widehat{P}_j Z_j$.*

---

**Algorithm 1** Basic label aggregation algorithm.

---

1: Form association matrices $X_j = Z_j^T Z_j$.
2: Form the average association matrix $\bar{X} = \sum_{j=1}^N w_j X_j$.
3: Perform $K$-means on the rows of $\bar{X}$.

---

**Algorithm 2** Spectral label aggregation algorithm.

---

1: Define the same average association matrix $\bar{X}$ as in Algorithm 1.
2: Perform $K$-truncated eigendecomposition of $\bar{X} = U\Lambda U^T$, where $\Lambda \in \mathbb{R}^{K \times K}$ contains top-$K$ eigenvalues on the diagonal and the columns of $U \in \mathbb{R}^{n \times K}$ are the corresponding eigenvectors.
3: Perform $K$-means on the rows of $U$.

---

## 2.1 LIFTING TO ASSOCIATION MATRICES

The difficulty in the unsupervised setting is that the optimal permutations $\{\widehat{P}_j\}$ are unknown. To get around this issue, we lift the barycenter problem to the space of association matrices. For a label matrix $Z \in \mathcal{E}_K^n$, we define the corresponding *association matrix* as $X = Z^T Z \in \{0,1\}^{n \times n}$. Note that $X_{ij} = 1$ iff $i$ and $j$ are in the same cluster according to $Z$, otherwise $X_{ij} = 0$. The advantage of $X$ is that it is invariant to label switching: $X = Z^T Z = (PZ)^T PZ$ for any permutation matrix $P$. This suggests solving the following *lifted* barycenter problem instead of (3):

$$\widehat{X} \in \underset{X \in \mathcal{X}_K}{\arg\min} \sum_{j=1}^N w_j \|X - X_j\|_F^2 \tag{6}$$

where $X_j = Z_j^T Z_j$ and $\mathcal{X}_K = \{Z^T Z : Z \in \mathcal{E}_K^n\}$, that is, the set of (binary) association matrices with at most $K$ clusters.

**Semidefinite relaxation**  Problem (6) is still hard to solve due to the combinatorial nature of $\mathcal{X}_K$. An approach to solving problems over $\mathcal{X}_K$ is to relax to a semidefinite program, an idea that has been applied before to community detection in networks (4). In particular, $\mathcal{X}_K$ is inside the doubly nonnegative cone $\{X : X \succeq 0, \ X \geq 0\}$, where $X \succeq 0$ and $X \geq 0$ mean $X$ is positive semidefinite and elementwise nonnegative. We note that $X_{ii} = 1$ for all $i$. This suggests relaxing to the following problem

$$\widehat{X} \in \underset{X}{\arg\min}\Big\{\sum_{j=1}^N w_j \|X - X_j\|_F^2 : \ X \succeq 0, \ X \geq 0, \ X_{ii} = 1\,\forall i\Big\}. \tag{7}$$

Problem (7) has a simple solution. The solution of the unconstrained version of (7) over $\mathbb{R}^{n \times n}$ is $X' := \sum_{j=1}^n w_j X_j / \sum_{j=1}^n w_j$. Since $\{X_j\}$ belong to the constraint set of (7) and this set is convex, $X'$ too belongs to the constraint set. Hence, $X'$ is the solution of (7), that is, $\widehat{X} = X'$. It remains to translate $\widehat{X}$ back to labels, for which we can preform rowwise $K$-means, leading to Algorithm 1.

Since elements of $\mathcal{X}_K$ are of rank at most $K$, to get a solution which is closer to that of the lifted barycenter problem (6), we can perform a spectral truncation of $\widehat{X}$ to its $K$ top eigenvectors, before applying the rowwise $K$-means. This leads to the spectral aggregation Algorithm 1. Other variants of spectral clustering on $\widehat{X}$ are also possible, e.g., using the normalized Laplacian, etc. In the $K$-means step of Algorithm 1 and 2, any constant-factor approximation to the $K$-means problem can be used.

## 2.2 CONSISTENCY

In order to study the statistical performance of different aggregation algorithms, we propose a *random perturbation model (RPM)*, where both the clusters and the labels of the clusters can undergo random perturbations, allowing us to study the difficulty of the unsupervised aggregation problem. Let $Z^* \in \mathcal{E}_K^n$ be the "true" label matrix with columns $z_i^* \in \mathcal{E}_K, i = 1, \ldots, n$.

**Definition 1** (RPM)**.** We write $Z \sim \mathcal{L}(Z^*, p)$ if $Z = (z_1, \ldots, z_n) \in \mathcal{E}_K^n$ with columns $z_i$ drawn independently as follows:

$$z_i = P z_i', \quad z_i' \sim (1-p)\delta_{z_i^*} + p\,\mathrm{Unif}(\mathcal{E}_K) \tag{8}$$

where $\delta_{z_i^*}$ is the point mass at $z_i^*$ and $P$ is an independent $K \times K$ permutation matrix.

Under RPM, the observed label matrices $Z_1, \ldots, Z_N$ are i.i.d., so it is reasonable to let $w_1 = \cdots = w_N = 1$ in Algorithms 1 and 2. We will discuss the algorithm for weighted samples in Section 5. Let $n_k(Z^*)$ be the number of objects in cluster $k$ according to $Z^*$. We make the following assumption:

$$n_k(Z^*) \leq \beta n/K, \quad k \in [K] \tag{9}$$

for some $\beta \in [1, K]$. Here, $\beta$ measures how much the true clustering deviates from being *balanced*. For $\beta = 1$, we have $n_k(Z^*) = n/K$ for all $k$, while $\beta = K$ corresponds to no restriction on the sizes of the true clusters. The following result shows that the basic aggregation algorithm is statistically consistent under the RPM:

**Theorem 1** (Consistency). *Let $Z^* \in \mathcal{E}_K^n$ be a label matrix satisfying (9). Assume that $Z_1, \ldots, Z_N$ are i.i.d. draws from $\mathcal{L}(Z^*, p)$ and let $\xi := p(2 - p)$. Let* Mis *be the misclassification rate between the true label matrix $Z^*$, and the output of Algorithm 1 with $w_j = 1$ for all $j \in [N]$. Then, there exists a universal constant $C > 0$, such that*

$$\mathbb{E}[\text{Mis}] \leq \frac{C\xi\beta^2 K}{(1-\xi)^2}\Big(\frac{2\beta^2}{N} + \frac{\xi K}{n}\Big). \tag{10}$$

Consistency of Algorithm 1 follows from (10) and the Markov inequality: For any $\delta > 0$, we have $\mathbb{P}(\text{Mis} \geq \delta) \leq \delta^{-1}\mathbb{E}[\text{Mis}] \to 0$ as $n, N \to \infty$ and $p$ is bounded away from 1. We note that in this and subsequent results all the parameters, such as $K$ and $p$, are allowed to change as $n, N \to \infty$, subject to the conditions of the theorems.

The first term inside the parentheses in (10) is the dominant one. Assume for simplicity that $\beta \asymp 1$. If the model has low noise, then $p$ is small and so is $\xi$ since $\xi \asymp p$. Then, the dominant term is $O(pK/N)$, that is, a smaller number of clusters, $K$, and a larger sample size, $N$, improve the performance. The second term is independent of of the sample size, but vanishes at the rate $O(p^2 K^2/n)$ in the low noise setting, as the number of objects, $n$, grows.

## 2.3 LITERATURE COMPARISON

The relation between the metrics on clusterings is discussed in (34; 33). Let $d'_M(Z, Z_j)$ be the Mirkin metric between clusterings $Z$ and $Z_j$ (34, Eqn (6)). As we show in Appendix B, it turns out that $d'_M(Z, Z_j) = \|X - X_j\|_F^2 = \|X - X_j\|_{\ell_1}$ leading to $d'_M/2 = \text{SDD} = \text{Binder} = \binom{n}{2}(1 - \text{Rand})$ where Binder denotes the Binder loss (5) and Rand, the Rand index (36). It follows that the lifted barycenter (6) that we derived is equivalent to the median partition (12) and the Binder loss barycenter of (45) as well as the Rand barycenter (15). This problem is often solved by greedy search starting from a random initialization (45; 8). Our Algorithm 2 then provides a fast scalable spectral method of obtaining an approximate solution to this ubiquitous problem. A lot of algorithms proposed for consensus clustering operate on the average association matrix $\bar{X}$, by treating it as a similarity matrix and performing usual clustering on it, for example, by performing agglomerative clustering (31; 18; 14; 28). Wade and Ghahramani (45) criticize these approaches as being ad-hoc compared to the decision-theoretic approach of finding a barycenter. However, we show $\bar{X}$ is indeed a solution to the relaxed version (7) of the barycenter (6) which is the same as the Binder barycenter in (45), hence clustering $\bar{X}$ is effectively solving the same problem with a different method.

A $k$-means based aggregation algorithm, called KCC, has been proposed in (48). In our notation, this is equivalent to concatenating $Z_j$s row-wise to form an $NK \times n$ matrix and running $k$-means on the columns. This is different from our Algorithm (1) that operates on $\bar{X}$. We compare with KCC in Section 4. Unlike KCC, Algorithm (1) comes with a consistency guarantee under our model assumptions. The BestOfK (12) essentially solves (6) by restricting the feasible region to $\{X_1, \ldots, X_N\}$, hence picking the lowest-scoring input clustering. The approach proposed in (7; 9) (see also (15)) is to find the input clustering that minimizes the cost $X \mapsto \|X - \bar{X}\|_F$. It is not hard to see that the barycenter cost (6) is equal to $\|X - \bar{X}\|_F^2$ plus a constant (essentially a bias-variance decomposition; see (26)). Hence, the approach of (7; 9) is equivalent to the BestOfK. As we show in Appendix A, BestOfK is, in general, inconsistent under RPM unless $N$ grows exponentially in $n$, a very strong condition not needed by our algorithms.

The $K$-means step 3 in Algorithm (1) can be replaced by other clustering algorithms, e.g. average-linkage clustering, BestOfK and CC pivot algorithm of (18; 12). Besides global clustering algorithms, many authors, including (41; 17; 18; 45; 8) also propose local search, a.k.a. greedy, algorithms which

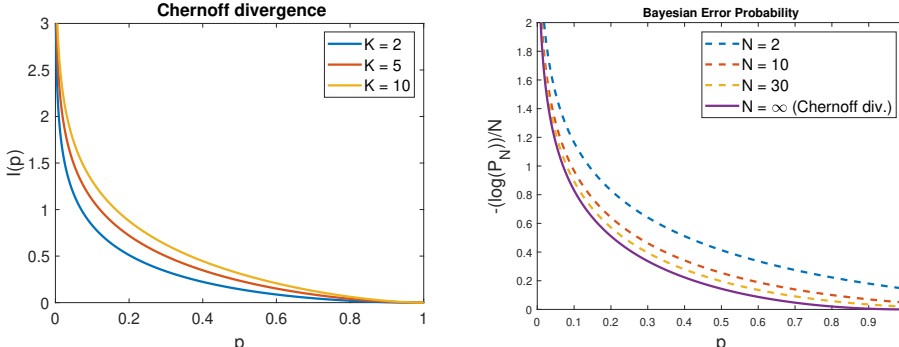

Figure 1: Left: Chernoff divergence $I$ in equation (11) as a function of $p$, the label perturbation probability. Right: How fast $-(\log P_N)/N$ converges to the Chernoff divergence for $K = 2$.

update one label at a time to minimize the barycenter loss, based on any number of metrics discussed earlier, including the information-theoretic ones (NMI and VI (32)). In (26), after reducing (6) to minimizing $X \mapsto \|X - \bar{X}\|_F^2$, they write $X = Z^T Z$ (in our notation) and then relax $Z$ to a general nonnegative matrix and solve the problem with nonnegative matrix factorization. Their work has the flavor of our Algorithm 2, although our approach, being based on regular spectral decomposition, is highly robust and scalable.

The RPM, with $P$ set equal to the identity, is closely related to the artificial data model considered in (18), with the difference that RPM does not potentially create a new cluster after perturbation, and introduces a random permutation of the clusters labels after perturbation (via $P$). In Appendix E, we argue that RPM is a good model of a concentrated posterior, hence consistency under RPM is relevant to Bayesian aggregation problems for which posterior consistency has been shown.

## 3 Optimal Rate

Theorem 1 guarantees an $O(N^{-1})$ rate of misclassification for Algorithm 1. A natural question is whether we can do better. To answer this question, we first consider what is the best an oracle, with the knowledge of the random permutations in (8), can do. This oracle is effectively solving the supervised version of the problem. We then show that a refinement step allows us to achieve nearly the same as the optimal oracle rate, without knowing the matching permutations.

### 3.1 Supervised oracle

Let $Z'$ be a label matrix with the $i$th column $z_i' \sim (1-p)\delta_{z_i^*} + p \, \mathrm{Unif}(\mathcal{E}_K)$, and let $Z_1, \ldots, Z_N$ be independent copies of $Z'$. We would like to recover the true label matrix $Z^*$. This is the oracle version of model (8), since $Z_j$s are label matrices without random permutations. In this case, the problem decouples to $n$ independent label recovery problems. We further simplify the problem to that of deciding between $z_1^* = e_1$ and $z_1^* = e_2$. This problem is equivalent the hypothesis testing:

$$H_0 : \mathrm{Multinomial}(N, (1-\tilde{p}, q, \ldots, q)) \quad \text{versus} \quad H_1 : \mathrm{Multinomial}(N, (q, 1-\tilde{p}, \ldots, q)),$$

where $q := p/K$ and $\tilde{p} := (K-1)q := p-q$. A classical result from information theory (6, Theorem 11.9.1) allows us to determine the optimal performance in this case:

**Proposition 2.** *The Bayesian error probability, $P_N$, for testing $H_0$ against $H_1$, with positive prior probabilities, is bounded by $e^{-NI}$, where*

$$I := -\log\big(2\sqrt{(1-\tilde{p})q} + (K-2)q\big) \tag{11}$$

*is the best achievable error exponent in the sense that $-\frac{1}{N}\log P_N \to I$ as $N \to \infty$.*

The left panel in Figure 1 shows plots of $I$ as a function of $p$, for various $K$, and the right panel shows the convergence of the exponent of $P_N$ for $K = 2$. The Bayesian error probability $P_N$ can be achieved by performing a likelihood ratio test between the two hypotheses. We can generalize this result to testing between $K$ hypotheses, in which case the Bayesian error probability is bounded by

---

**Algorithm 3** Local Refinement

---

1: **Input:** Average association matrix $\bar{X}$, initial label matrix $\widetilde{Z}$.
2: **Output**: An updated label matrix $\widehat{Z}$.
3: **for** $i = 1$ to $N$ **do**
4:     Let $\bar{X}_i$ be the $i$th column of $\bar{X} = \sum_j w_j X_j$.
5:     Replace the $i$th column of $\widetilde{Z}$ by zeros and denote this matrix by $\widetilde{Z}_{-i}$.
6:     Let $(n_1, \ldots, n_K)$ be the row sums of $\widetilde{Z}_{-i}$.
7:     Let $(b_1, \ldots, b_K) = \widetilde{Z}_{-i} \bar{X}_i$.
8:     Update the $i$th label by $\arg \max_k (b_k/n_k)$.
9: **end for**

---

$(K - 1)e^{-NI}$. The oracle algorithm that achieves this bound is the one that finds the columnwise "argmax" of the average of $Z_j$s. In light of Proposition 2, the bound in Theorem 1 is far from optimal since it guarantees a linear decay of the error in $N$, that is $O(N^{-1})$, rather than the exponential decay $e^{-NI}$. The question is whether this gap can be filled by a non-oracle algorithm.

## 3.2    LOCAL REFINEMENT

To approach the oracle rate, we propose a fast local refinement on the label of each object, as outlined in Algorithm 3. This refinement can be performed on the output of any reasonable aggregation algorithm. The idea of performing a local refinement to boost the performance of clustering algorithms has been employed in various settings, including clustering of sub-Gaussian mixtures (30) and community detection in stochastic block models (SBM) (3; 16; 50; 51). A more detailed comparison with the SBM appears in Appendix F.

Algorithm 3 requires a good initial label matrix $\widetilde{Z}$, with a small number of mismatches relative to the true matrix $Z^*$. The algorithm focuses on the local information of the $i$th object. With $\bar{X}$ the average of the association matrices, $\bar{X}_{ij}$ is the sample proportion of objects $i$ and $j$ appearing in the same cluster. Viewing $\bar{X}$ as the adjacency matrix of a weighted graph, one can verify that $b_k = \sum_{j \neq i} w_j X_{ji} 1\{\widetilde{Z}_j = k\}$ is the weighted number of connections between object $i$ and objects in cluster $k$. The algorithm then normalizes the number of connections by the cluster size $n_k$. Thus, $b_k/n_k$ estimates the probability that object $i$ is connected to another object in cluster $k$. The higher this probability is, the higher the chance that object $i$ belongs to cluster $k$. The last step in the for loop, updates the $i$th label according to these statistics.

It is possible to repeat the local refinement procedure, by feeding its output $\widehat{Z}$ back as an initial label matrix. Given a good initialization, local refinements usually converge in constant or $O(\log n)$ number of steps (30).

## 3.3    ACHIEVING THE SUPERVISED RATE WITHOUT SUPERVISION

From the oracle result (Proposition 2), we expect the optimal misclassification rate to be close to $e^{-NI}$. This is verified by the next result, showing that a single local refinement step applied to a consistent aggregation algorithm, such as Algorithm 1, can get us nearly to the optimal rate:

**Theorem 2.** *Assume that $Z_1, \ldots, Z_N$ are i.i.d. draws from the random perturbation model (8). Let $\widehat{Z}$ be the output of Algorithm 3 initialized by some, possibly data-dependent, $\widetilde{Z}$. Let $n_{min} = \min_{k \in [K]} n_k(Z^*)$ be the smallest true cluster size and recall the definition of $I$ in (11). Assume that*

*(a)  $n_{min}p(1 \wedge I)/K \to \infty$ and $\frac{NI}{\log K} \to \infty$,*

*and there exists $\delta$ satisfying $Kn\delta/(n_{min}pI) = o(1)$ such that one of the followings holds:*

*(b1)  $\mathbb{P}(\text{Mis}(\widetilde{Z}, Z^*) \leq \delta) = 1 - o(1)$,     or     (b2)  $\mathbb{E}[\text{Mis}(\widetilde{Z}, Z^*)] \leq \delta$.*

*Then, for some $\eta = o(1)$, the misclassification rate satisfies*

$$\mathbb{P}\big( \text{Mis}(\widehat{Z}, Z^*) \leq e^{-(1-\eta)NI} \big) = 1 - o(1). \tag{12}$$

The assumptions of Theorem 2 are mild. Suppose that $p$ is bounded away from 0 or 1, say $p \in [0.01, 0.99]$, $K = O(1)$ and the cluster sizes are similar. Then, the assumptions can be simplified to $n_{\min}, N \to \infty$ and $\delta = o(1)$. The theorem is most interesting when $p \to 1$ and $K$ is large. In this case, $I \to 0$ and the first requirement of assumption (a) becomes $n_{\min} I/K \to \infty$. This assumption guarantees that the samples provide sufficient information to recover the permutations, although we have not attempted to do so in our algorithm. The second requirement of assumption (a) provides evidence to distinguish the true labels from the other $K - 1$ labels.

Under the assumptions of Theorem 2, Algorithm 3 initialized with input satisfying (12), will have an output satisfying (b1). Hence, Theorem 2 also guarantees rate-optimality of an iterative Algorithm 3.

## 4 EXPERIMENTAL RESULTS

We now present empirical results comparing the performances of the proposed aggregation algorithms, with additional results provided in Appendix C. The ground truth label matrix $Z^*$ is generated by randomly assigning each of the $n$ objects to one of the $K$ labels. The $N$ input clusterings $Z_j, j \in [N]$ are generated from model (8). We measure the performance of an algorithm by the *adjusted Rand index (ARI)* of its output against the ground truth. We consider seven different aggregation algorithms: (1) Our Algorithm 1, referred to as "Basic" in the plots; (2) Our Algorithm 2, referred to as SC; (3) KCC algorithm (48); (4) CC Pivot algorithm (2; 18) with threshold 0.25; (5) Best One Element Move (BOEM) algorithm (11; 18); (6) the EM algorithm of (24); and (7) BestOfK algorithm (12; 18). In addition, we consider variants of algorithms (1)–(5) where we apply our refinement step to their output. This gives us a total of 12 methods. In the plots, the refined version is denoted with a solid line and the original version (without refinement) with a dotted line. We also use the average ARI of the $N$ input clusterings, denoted by the INPUT, as a baseline.

**Balanced setting with varying $n$ and $N$.** Figure 2 depicts plots of ARI versus the noise probability $p$ in model (8), for various methods. The results are averaged over 40 replications. The settings in Figure 2 all correspond to balanced cluster sizes. Generally, our proposed Basic and SC algorithms outperform the EM, KCC, CCPivot and BOEM algorithms, with the failure thresholds occurring at larger values of $p$ (harder problems). In some settings, the refinement shows some improvement, but in others, the output of the refinement applied to Basic and SC nearly coincides with the original algorithm. This shows that in some settings the original algorithm implicitly performs the refinement itself. We note that increasing $N$ shifts the failure thresholds to the right as expected, as does the increasing of $n$, both consistent with our theory. Note that BestOfK performs no better than INPUT. Moreover, refined versions of KCC and BOEM outperform their original versions.

**Unbalanced setting.** Figure 3 depicts the results obtained with disproportionate cluster sizes, specifically with $p_1$ proportion of the objects in one cluster and the rest distributed uniformly to the remaining clusters. Local refinement over Basic and SC performs significantly better as we deal with input clusterings of disproportionate cluster sizes, especially at lower noise probabilities.

## 5 CONCLUSION AND DISCUSSION

In the present paper, we defined the random perturbation model to study the label aggregation problem. We developed a $K$-means type algorithm followed by an efficient local refinement step to achieve the optimal misclassification rate under the assumptions of the model. Numerical experiments also show the effectiveness of our proposed methods. Let us also discuss possible avenues for future work.

**A single-stage algorithm.** Two-stage algorithms have been popular in many clustering problems (3; 30; 16). Numerical experiments show that, in many cases, a $K$-means type algorithm performs sufficiently close to the local refinement with good initialization (Section 4). The $K$-means type algorithm assigns labels based on the distances between the objects and the centers. This criterion is different from the likelihood ratio test in many cases, and so the output of the algorithm will not achieve the misclassification rate of the oracle problem. It will be a novel improvement if the $K$-means type algorithm can be generalized to an EM-type algorithm so that the "distance" between the parameter and the object is defined by the likelihood. Whether such an algorithm exists, and how efficient it is statistically and computationally, are interesting open questions.

**A robust algorithm.** We observe i.i.d. label matrices from the random perturbation model defined in Definition 1. As long as $p < 1$, every label matrix from this model provides the same amount

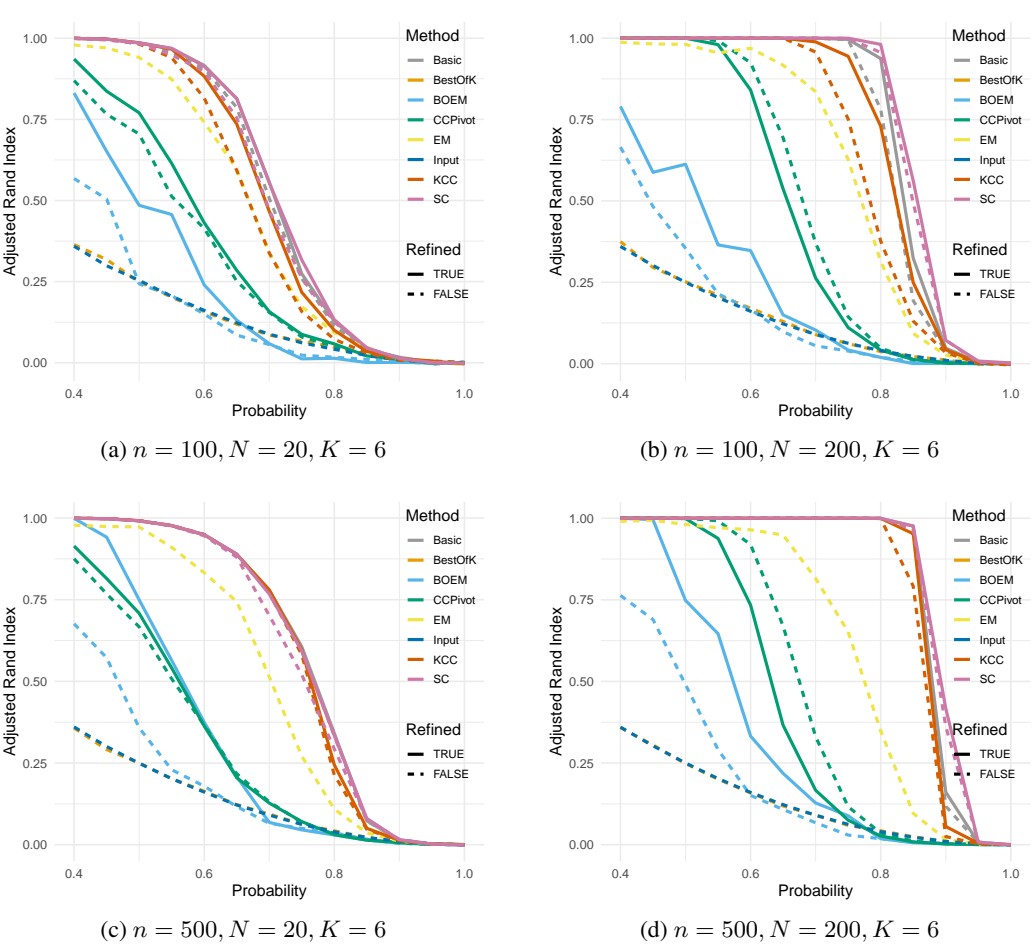

Figure 2: Performance impact with the increase in $n$ and $N$

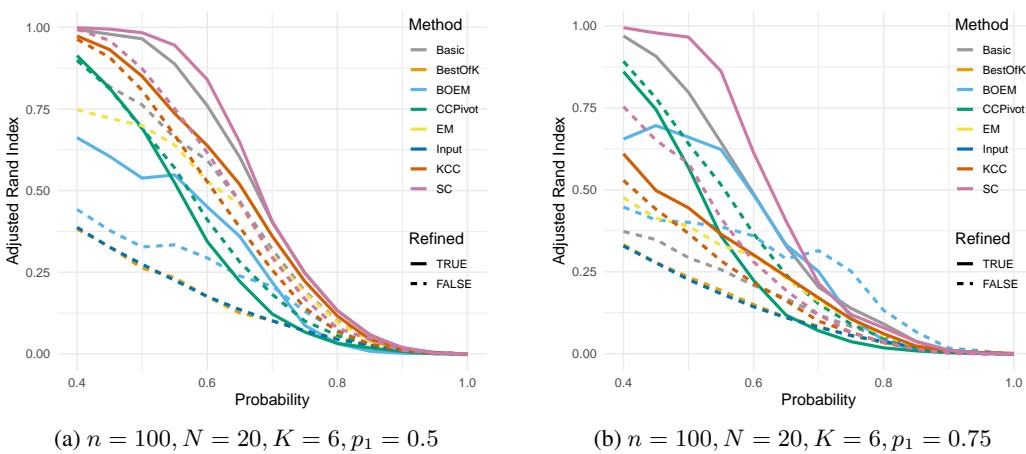

Figure 3: Significant improvements due to local refinement in the case of unbalanced cluster sizes.

of information in expectation, so there is no reason to assign different weights to label matrices. However, in practice, the samples may not be i.i.d.

## ACKNOWLEDGEMENT

Zhixin Zhou was supported by the Research Grants Council of Hong Kong [Project Number 21502921]. A. A. Amini was supported by the NSF grant DMS-1945667.

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

# Supplementary material for
# "Statistical Guarantees for Consensus Clustering"

This supplement contains the detailed proofs of the results and some extra simulations.

## A    INCONSISTENCY OF BESTOFK

Using the notation of the present paper, the name "BestOfK" should be "BestOfN". We will use our notation in the following proposition and keep the name "BestOfK".

**Proposition 3.** *BestOfK is not consistent unless $N$ grows exponentially fast in $n$.*

*Proof.* We will prove this proposition by providing a counterexample. Suppose $K = 2$, $1 - \tilde{p} = 0.6$ and $q = 0.4$. Then for a label vector $z$ from the RPM, by the Hoeffding inequality,

$$\mathbb{P}(\text{Mis}(z, z^*) \geq 0.1) \geq 1 - \exp(2(0.4 - 0.1)^2 n) - \exp(2(0.6 - 0.1)^2 n) \geq 1 - 2\exp(-0.18n)$$

where we have accounted for the two permutations in the definition of Mis. Suppose we observe $N$ i.i.d. label vectors $z_1, \ldots, z_N$ from the RPM. Then

$$\mathbb{P}(\min_{i \in [N]} \text{Mis}(z_i, z^*) \geq 0.1) \geq (1 - 2\exp(-0.18n))^N \geq 1 - 2N\exp(-0.18n).$$

This probability (of missing the target) approaches 1 unless $N$ grows exponentially fast in $n$.    □

## B    RELATIONS AMONG CLUSTERING DISTANCES

Let $n_k$ be the size of the $k$th cluster of $Z \in \mathcal{E}_K^n$ and, $n_\ell^*$ the size of the $\ell$th cluster of $Z^* \in \mathcal{E}_L^n$, and let $X$ and $X^*$ be the corresponding association matrices. The Mirkin distance (34, Eqn (6)) is given by

$$d'_M(Z, Z^*) = \sum_k n_k^2 + \sum_\ell (n_\ell^*)^2 - 2 \sum_{k,\ell} n_{k\ell}^2 \tag{13}$$

where $n_{k\ell}$ is the number of objects that are in cluster $k$ according to $Z$ and cluster $\ell$ according to $Z^*$. It is not hard to see that $\sum_k n_k^2 = \|X\|_F^2$ and similarly $\sum_\ell (n_\ell^*)^2 = \|X^*\|_F^2$. We also have $Z(Z^*)^T = (n_{k\ell})$, hence, using $\|A\|_F^2 = \text{tr}(AA^T)$,

$$\sum_{k,\ell} n_{k\ell}^2 = \|Z(Z^*)^T\|_F^2 = \text{tr}(Z(Z^*)^T Z^* Z^T) = \text{tr}((Z^*)^T Z^* Z^T Z) = \text{tr}(X^* X).$$

Combining these facts, we obtain the first equality below

$$d'_M(Z, Z^*) = \|X - X^*\|_F^2 = \|X - X^*\|_{\ell_1}. \tag{14}$$

The second equality follows from $X - X^*$ having elements in $\{-1, 0, 1\}$. Here $\|\cdot\|_{\ell_1}$ denotes the $\ell_1$ norm of a matrix viewed as a vector. The equality $d'_M(Z, Z^*) = \|X - X^*\|_{\ell_1}$ immediately shows that $d'_M$ is indeed a distance on the space of clusterings. It also connects the Mirkin distance with the Rand index.

To see the connection with the Rand index, let $N_{\text{disagree}}$ be the number of pairs of objects for which $Z$ and $Z'$ disagree about their co-clustering, that is, whether the two objects are in the same cluster or not. Similarly, let $N_{\text{agree}}$ be the number of pairs of objects for which $Z$ and $Z'$ agree about their co-clustering. We have $N_{\text{disagree}} + N_{\text{agree}} = \binom{n}{2}$. The Rand index is defined as the proportion of the agreements, that is,

$$\text{Rand} = \frac{N_{\text{agree}}}{\binom{n}{2}}.$$

It is easy to see that $\|X - X^*\|_{\ell_1} = 2N_{\text{disagree}}$ where the factor of 2 is due to the double-counting caused by the symmetry of $X - X^*$. This proves the relation

$$\frac{1}{2} d'_M = \binom{n}{2}(1 - \text{Rand}). \tag{15}$$

The symmetric difference distance (SDD) is another name for $N_{\text{disagree}}$, hence $d'_M/2 = \text{SDD}$. The Binder loss is defined as half the expression in (13), that is, $d'_M/2 = \text{Binder}$.

### B.1 Consistency in Mis implies consistency in Mirkin distance

Let us now show that the consistency in Mis implies consistency in the *normalized* Mirkin distance defined as $d_M := d'_M/n^2$. See (34, Eqn (9)). It then follows that consistency in Mis implies consistency in the normalized SDD, normalized Binder loss and the Rand index, as discussed above. This claim follows from the following inequality:

**Proposition 4.** *We have $d_M \leq 2 \cdot \mathrm{Mis}$.*

*Proof.* Let $X$ and $X^*$ be the association matrices corresponding to label vectors $z$ and $z^*$. Then $d'_M(z, z^*) = \|X - X^*\|_{\ell^1}$ as shown in (14). The entries of $X - X^*$ take values in $\{-1, 0, 1\}$. Assume, WLOG, that the optimal permutation between $z$ and $z^*$ is the identity. Then:

1. If the label $z_i = z_i^*$, then the $i$th row of $X - X^*$ has at most "$n \cdot \mathrm{Mis}$" nonzero entries. There are at most $n$ such rows.

2. If the label $z_i \neq z_i^*$, then the $i$th row of $X - X^*$ has at most $n$ nonzero entries. There are at most "$n \cdot \mathrm{Mis}$" such rows.

Therefore, $d'_M = \|X - X^*\|_{\ell^1} \leq n \cdot (n \cdot \mathrm{Mis}) + (n \cdot \mathrm{Mis}) \cdot n = 2n^2 \cdot \mathrm{Mis}$ and the result follows. $\square$

## C Extra Simulation Results

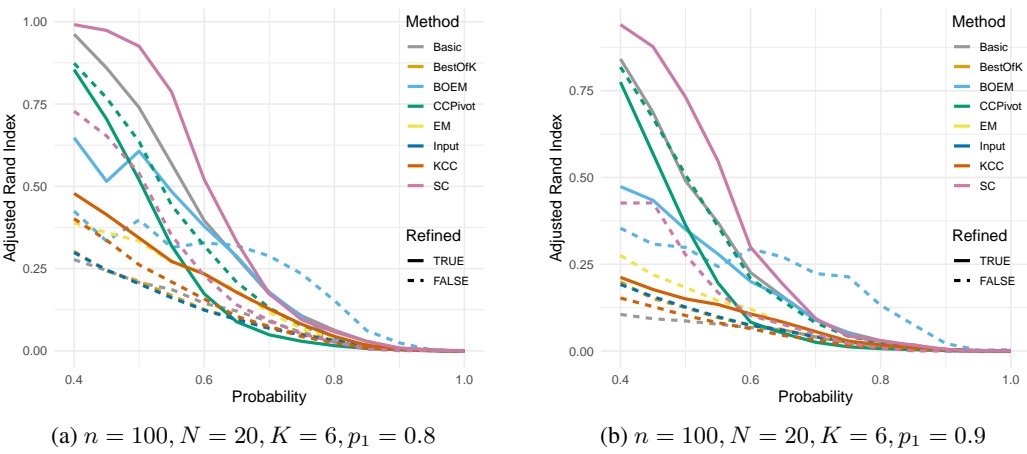

(a) $n = 100, N = 20, K = 6, p_1 = 0.8$        (b) $n = 100, N = 20, K = 6, p_1 = 0.9$

Figure 4: Significant improvements due to local refinement in the case of unbalanced cluster sizes.

Figure 4 shows some extra cases of unbalanced cluster sizes (various values of $p_1$ as defined earlier), showing the significant improvement of the refinement step in such cases. All the results for the unbalanced case (including those in the main text) are averaged over 120 runs.

Tables 1, 2 and 3 show the average ARI in all the eight settings (abbreviated Set in the tables) shown in Figures 2, 3 and 4. The tables show the performance of the methods at noise probabilities $p = 0.45, 0.55$ and $0.65$ respectively—corresponding to a cross-section of each plot at a line parallel to the $y$-axis, crossing the $x$-axis at the respective value of $p$. The settings are as follows:

1. Set 1: Balanced, $n = 100, N = 20$.

2. Set 2: Balanced, $n = 100, N = 200$.

3. Set 3: Balanced, $n = 500, N = 20$.

4. Set 4: Balanced, $n = 500, N = 200$.

5. Set 5: Unbalanced, $n = 100, N = 20, p_1 = 0.5$.

6. Set 5: Unbalanced, $n = 100, N = 20, p_1 = 0.75$.

| Method | Refined | Set 1 | Set 2 | Set 3 | Set 4 | Set 5 | Set 6 | Set 7 | Set 8 |
|---|---|---|---|---|---|---|---|---|---|
| Basic | FALSE | 1.00 | 1.00 | 1.00 | 1.00 | 0.82 | 0.35 | 0.24 | 0.093 |
| Basic | TRUE | 1.00 | 1.00 | 1.00 | 1.00 | 0.98 | 0.91 | 0.86 | 0.690 |
| BestOfK | FALSE | 0.32 | 0.30 | 0.29 | 0.31 | 0.33 | 0.28 | 0.24 | 0.150 |
| BOEM | FALSE | 0.50 | 0.48 | 0.57 | 0.69 | 0.38 | 0.41 | 0.33 | 0.310 |
| BOEM | TRUE | 0.65 | 0.59 | 0.94 | 1.00 | 0.60 | 0.70 | 0.52 | 0.430 |
| CCPivot | FALSE | 0.77 | 1.00 | 0.77 | 1.00 | 0.81 | 0.78 | 0.77 | 0.670 |
| CCPivot | TRUE | 0.84 | 1.00 | 0.81 | 1.00 | 0.81 | 0.75 | 0.70 | 0.570 |
| EM | FALSE | 0.97 | 0.98 | 0.97 | 0.99 | 0.72 | 0.41 | 0.36 | 0.220 |
| Input | FALSE | 0.30 | 0.30 | 0.30 | 0.30 | 0.33 | 0.28 | 0.25 | 0.160 |
| KCC | FALSE | 1.00 | 1.00 | 1.00 | 1.00 | 0.91 | 0.44 | 0.34 | 0.130 |
| KCC | TRUE | 1.00 | 1.00 | 1.00 | 1.00 | 0.93 | 0.50 | 0.41 | 0.180 |
| SC | FALSE | 0.99 | 1.00 | 1.00 | 1.00 | 0.96 | 0.65 | 0.65 | 0.430 |
| SC | TRUE | 1.00 | 1.00 | 1.00 | 1.00 | 0.99 | 0.98 | 0.97 | 0.880 |

Table 1: Mean adjusted rand index (ARI) for all settings at noise probability $p = 0.45$.

| Method | Refined | Set 1 | Set 2 | Set 3 | Set 4 | Set 5 | Set 6 | Set 7 | Set 8 |
|---|---|---|---|---|---|---|---|---|---|
| Basic | FALSE | 0.96 | 1.00 | 0.98 | 1.00 | 0.66 | 0.26 | 0.19 | 0.077 |
| Basic | TRUE | 0.97 | 1.00 | 0.98 | 1.00 | 0.89 | 0.64 | 0.57 | 0.370 |
| BestOfK | FALSE | 0.20 | 0.21 | 0.20 | 0.20 | 0.23 | 0.19 | 0.17 | 0.096 |
| BOEM | FALSE | 0.21 | 0.21 | 0.23 | 0.29 | 0.33 | 0.39 | 0.31 | 0.240 |
| BOEM | TRUE | 0.46 | 0.36 | 0.56 | 0.65 | 0.55 | 0.62 | 0.48 | 0.280 |
| CCPivot | FALSE | 0.51 | 0.99 | 0.51 | 0.99 | 0.57 | 0.52 | 0.44 | 0.360 |
| CCPivot | TRUE | 0.61 | 0.98 | 0.54 | 0.94 | 0.52 | 0.36 | 0.32 | 0.200 |
| EM | FALSE | 0.87 | 0.95 | 0.91 | 0.97 | 0.64 | 0.33 | 0.27 | 0.140 |
| Input | FALSE | 0.20 | 0.20 | 0.20 | 0.20 | 0.23 | 0.18 | 0.16 | 0.098 |
| KCC | FALSE | 0.94 | 1.00 | 0.98 | 1.00 | 0.67 | 0.28 | 0.21 | 0.081 |
| KCC | TRUE | 0.97 | 1.00 | 0.98 | 1.00 | 0.74 | 0.36 | 0.27 | 0.130 |
| SC | FALSE | 0.95 | 1.00 | 0.98 | 1.00 | 0.75 | 0.41 | 0.35 | 0.170 |
| SC | TRUE | 0.97 | 1.00 | 0.98 | 1.00 | 0.95 | 0.86 | 0.79 | 0.550 |

Table 2: Mean adjusted rand index (ARI) for all settings at noise probability $p = 0.55$.

7. Set 6: Unbalanced, $n = 100$, $N = 20$, $p_1 = 0.8$.

8. Set 7: Unbalanced, $n = 100$, $N = 20$, $p_1 = 0.9$.

## D  PROOFS

### D.1  PROOF OF PROPOSITION 1

For any $Z \in \mathcal{E}_K^n$, we have $\|Z\|_F^2 = \sum_{k,i} Z_{ki}^2 = \sum_{k,i} Z_{ki} = n$. Thus, $\|Z\|_F^2 = \|\widehat{P}_j Z_j\|_F^2 = n$ for all $j \in [N]$. Hence, solving (5) is equivalent to maximizing $f(Z) := \sum_{j=1}^N w_j \, \mathrm{tr}(Z^T \widehat{P}_j Z_j) = \mathrm{tr}(Z^T \bar{Z})$ over $\mathcal{E}_K^n$, where $\bar{Z} := \sum_j w_j \widehat{P}_j Z_j$. Let $Z = (z_1, \ldots, z_n)$ and $\bar{Z} = (\bar{z}_1, \ldots, \bar{z}_n)$. Maximizing $f(Z) = \sum_{i=1}^n \langle z_i, \bar{z}_i \rangle$ is a separable problem over $i$, and maximizing $z \mapsto \langle z, \bar{z}_i \rangle$ over $\mathcal{E}_K$ amounts to finding the index of the maximum element of $\bar{z}_i$, that is, the "argmax" of $\bar{z}_i$, as claimed.

### D.2  PROOF OF THEOREM 1

We have $Z_j = P_j Z_j'$ where $Z_j' = (z_{j1}', \ldots, z_{jn}')$ and $z_{ji}'$ are i.i.d. draws as in (8). Since the algorithm is invariant to permutations $P_j$, without loss of generality we assume $P_j = I_n$, hence $Z_j = Z_j'$. We write $X^* = (Z^*)^T Z^*$ for the true association matrix. Let $E_n$ be the all-ones $n \times n$ matrix.

| Method | Refined | Set 1 | Set 2 | Set 3 | Set 4 | Set 5 | Set 6 | Set 7 | Set 8 |
|---|---|---|---|---|---|---|---|---|---|
| Basic | FALSE | 0.790 | 1.000 | 0.88 | 1.00 | 0.47 | 0.17 | 0.120 | 0.061 |
| Basic | TRUE | 0.810 | 1.000 | 0.89 | 1.00 | 0.60 | 0.33 | 0.280 | 0.160 |
| BestOfK | FALSE | 0.120 | 0.130 | 0.12 | 0.12 | 0.13 | 0.11 | 0.098 | 0.062 |
| BOEM | FALSE | 0.085 | 0.098 | 0.12 | 0.11 | 0.24 | 0.29 | 0.320 | 0.270 |
| BOEM | TRUE | 0.130 | 0.150 | 0.21 | 0.22 | 0.36 | 0.33 | 0.290 | 0.150 |
| CCPivot | FALSE | 0.250 | 0.690 | 0.22 | 0.67 | 0.28 | 0.24 | 0.210 | 0.140 |
| CCPivot | TRUE | 0.280 | 0.540 | 0.21 | 0.37 | 0.22 | 0.12 | 0.088 | 0.051 |
| EM | FALSE | 0.600 | 0.920 | 0.74 | 0.95 | 0.46 | 0.23 | 0.180 | 0.078 |
| Input | FALSE | 0.120 | 0.120 | 0.12 | 0.12 | 0.14 | 0.11 | 0.096 | 0.059 |
| KCC | FALSE | 0.590 | 1.000 | 0.89 | 1.00 | 0.39 | 0.16 | 0.100 | 0.045 |
| KCC | TRUE | 0.740 | 1.000 | 0.89 | 1.00 | 0.52 | 0.24 | 0.180 | 0.083 |
| SC | FALSE | 0.750 | 1.000 | 0.88 | 1.00 | 0.47 | 0.19 | 0.140 | 0.076 |
| SC | TRUE | 0.810 | 1.000 | 0.89 | 1.00 | 0.65 | 0.40 | 0.330 | 0.190 |

Table 3: Mean adjusted rand index (ARI) for all settings at noise probability $p = 0.65$.

**Lemma 1.** *Let $Z \sim \mathcal{L}(Z^*, p)$ and let $X = Z^T Z$ be the corresponding association matrix. Then,*

$$M := \mathbb{E}[X] = (1 - \xi)X^* + \xi\Big(\frac{1}{K}E_n + (1 - \frac{1}{K})I_n\Big) \tag{16}$$

*where $\xi = p(2 - p)$.*

*Proof of Lemma 1.* We have $X_{ij} = (Z^T Z)_{ij} = \langle z_i, z_j \rangle$ and $\mathbb{E}[z_i] = (1 - p)z_i^* + p\frac{1}{K}1_K$. For $i \neq j$, $z_i$ and $z_j$ are independent, hence

$$\mathbb{E}X_{ij} = \langle \mathbb{E}z_i, \mathbb{E}z_j \rangle = \langle (1 - p)z_i^* + p\frac{1}{K}1_K, (1 - p)z_j^* + p\frac{1}{K}1_K \rangle$$

$$= (1 - p)^2\langle z_i^*, z_j^* \rangle + 2p(1 - p)\frac{1}{K} + p^2\frac{1}{K}$$

For $i = j$, we have $\mathbb{E}[X_{ii}] = 1$. The above shows that

$$\mathbb{E}[X] = (1 - p)^2 X^* + p(2 - p)\frac{1}{K}E_n + p(2 - p)\Big(1 - \frac{1}{K}\Big)I_n$$

which simplifies to the desired expression. $\square$

Let $Z_1, \ldots, Z_N, Z \sim \mathcal{L}(Z^*, p)$ be independent draws, and let $X_j = Z_j^T Z_j$ and $X = Z^T Z$ be the associated association matrices. Setting $\bar{X} = \frac{1}{N}\sum_{t=1}^{N} X_t$, we obtain

$$\mathbb{E}\|\bar{X} - M\|_F^2 = \sum_{ij} \mathbb{E}(\bar{X}_{ij} - M_{ij})^2 = \sum_{ij} \text{var}(\bar{X}_{ij}) = \frac{1}{N}\sum_{ij} \text{var}(X_{ij}).$$

We have $\text{var}(X_{ij}) = 0$ for $i = j$. For $i \neq j$, one has $X_{ij} \sim \text{Ber}((1 - \xi)X_{ij}^* + \xi/K)$, hence

$$\text{var}(X_{ij}) = (1 - \xi)X_{ij}^* + \frac{\xi}{K} - \Big((1 - \xi)^2 X_{ij}^* + 2\frac{\xi}{K}(1 - \xi)X_{ij}^* + \frac{\xi^2}{K^2}\Big)$$

$$= \psi(\xi)\Big(1 - \frac{2}{K}\Big)X_{ij}^* + \psi(\xi/K)$$

where $\psi(x) = x(1 - x)$. Note that $\xi = p(2 - p) \in (0, 1)$. It follows that

$$N \cdot \mathbb{E}\|\bar{X} - M\|_F^2 \leq \psi(\xi)\Big(1 - \frac{2}{K}\Big)\sum_{ij} X_{ij}^* + n^2\psi(\xi/K)$$

where the inequality is due to bounding $\text{var}(X_{ii})$ by the same formula used for $\text{var}(X_{ij}), i \neq j$. Let $n_k^*$ be the number of entities in cluster $k$ of $Z^*$, that is, $n_k^* = (Z^*1_n)_k$. We have $\sum_{ij} X_{ij}^* = \|Z^*1_n\|^2 = \sum_k (n_k^*)^2$. Using the assumption $n_k^* \leq \beta n/K$, we have

$$N \cdot \mathbb{E}\|\bar{X} - M\|_F^2 \leq \psi(\xi)\Big(1 - \frac{2}{K}\Big)\frac{\beta^2 n^2}{K} + n^2\psi(\xi/K).$$

**Calculating the center separations.** Let $\widetilde{M} = (1 - \xi)X^* + (\xi/K)E_n$. We note that $M - \widetilde{M}$ is diagonal and

$$\|M - \widetilde{M}\|_F^2 = \|\xi(1 - 1/K)I_n\|_F^2 = \xi^2(1 - 1/K)^2 n \le \xi^2 n.$$

It follows that

$$\mathbb{E}\|\bar{X} - \widetilde{M}\|_F^2 = \mathbb{E}\sum_{i \ne j}(\bar{X}_{ij} - \widetilde{M}_{ij})^2 + \mathbb{E}\sum_i (\bar{X}_{ii} - \widetilde{M}_{ii})^2$$
$$\le \mathbb{E}\|\bar{X} - M\|_F^2 + \xi^2 n.$$

We obtain

$$\frac{1}{n^2}\mathbb{E}\|\bar{X} - \widetilde{M}\|_F^2 \le \frac{2}{N}\Big[\psi(\xi)\Big(1 - \frac{2}{K}\Big)\frac{\beta^2}{K} + \psi(\xi/K)\Big] + \frac{2\xi^2}{n}.$$

The matrix $\widetilde{M}$ is a $K$-means matrix with $K$ distinct rows. If $z_i = r \ne k = z_{i'}$, then

$$\|\widetilde{M}_{i*} - M_{i'*}\|^2 = (1 - \xi)^2\|X_{i*}^* - X_{i'*}^*\|^2 = (1 - \xi)^2(n_r^* + n_k^*) \ge 2(1 - \xi)^2\frac{n}{\beta K}$$

using $n_k^* \ge n/(\beta K)$, which holds by assumption (9). We have $n_r \delta_r^2 \ge 2(1 - \xi)^2(\frac{n}{\beta K})^2$ which gives the following bound, using (49, Proposition 1),

$$\mathbb{E}[\text{Mis}_r] \lesssim \frac{1}{N(1 - \xi)^2}\Big[\psi(\xi)(K - 2)\beta^4 + \beta^2 K^2 \psi(\xi/K)\Big] + \frac{\xi^2}{(1 - \xi)^2}\frac{\beta^2 K^2}{n}.$$

Here, $\text{Mis}_r$ is the misclassification rate over true cluster $r$. The dependence on $\beta$ of the first term is $O(\beta^2)$ when $K = 2$ and $O(\beta^4)$ when $K > 2$. Ignoring this difference, we can simplify the bound, by noting that $K^2\psi(\xi/K) = K\xi(1 - \xi/K) \le K\xi$ and $\beta^2 \le \beta^4$. Then,

$$\mathbb{E}[\text{Mis}_r] \lesssim \frac{\xi}{(1 - \xi)^2}\frac{2K\beta^4}{N} + \frac{\xi^2}{(1 - \xi)^2}\frac{\beta^2 K^2}{n},$$

from which the bound in the theorem follows since $\text{Mis} = \sum_r(n_r^*/n)\text{Mis}_r$.

### D.3 PROOF SKETCH FOR THEOREM 2

For the benefit of the readers, we first give a proof sketch for Theorem 2 and its key lemma. A detailed proof is given in Appendix D.4. The proof of Theorem 2 relies on the following key lemma:

**Lemma 2.** *Let $B(\delta)$ denote the set of label matrices $Z$ with at most $n\delta$ labels different from $Z^*$, and let $\widehat{Z}(Z)$ be the output of Algorithm 3 with initial label matrix $Z$. If $n_{min}p(1 \wedge I)/K \to \infty$ and $\frac{\log K}{NI} \to 0$, then*

$$\mathbb{P}\big(\exists Z \in B(\delta) \text{ such that } \widehat{z}_i(Z) \ne z_i^*\big) \le e^{-(1-\eta')NI + \frac{3Kn\delta N}{2pn_{min}}} \tag{17}$$

*for some $\eta' = o(1)$.*

The first step is to prove the case $\delta = 0$ in Lemma 2, corresponding to the initial label matrix in Algorithm 3 being $Z^*$. If $z_i^* = e_1$, then the algorithm fails to recover $z_i^*$ if there exists $k \ne 1$ such that $Y_k := n_1 b_k - n_k b_1 \ge 0$. $Y_k$ is the average of i.i.d. samples, where each sample follows a mixture model depending on which events among $z_i = e_1$, $z_i = e_k$, or $z_i \notin \{e_1, e_k\}$ happens. We compute the MGF of $Y_k$ and obtain the bound

$$\mathbb{E}\big[\exp(tNY_k/(n_1 n_2(1 - p)))\big] \le \big[(1 - \tilde{p})e^{-t(1+o(1))} + qe^{t(1+o(1))} + (K - 2)qe^{\frac{2qt^2}{n_{\min}(1-p)^2}}\big]^N.$$

The choice of $t$ has little affect on the last term since $n_{\min}$ is large, so we set $t = \frac{1}{2}\log[(1 - \tilde{p})/q]$ to minimize $(1 - \tilde{p})e^{-t} + qe^t$. Under the regularity conditions of the lemma and the definition of $I$ in (11), we have

$$\mathbb{E}\big[\exp(tY_k/(n_1 n_2(1 - p)))\big] \le \big[2\sqrt{(1 - \tilde{p})q} + (K - 2)q\big]^{(1-o(1))N} = e^{-(1-o(1))NI}.$$

Applying the Chernoff inequality, it follows that

$$\mathbb{P}(\widehat{z}_i(Z^*) \neq z_i^*) \leq \sum_{k=2}^{K} \mathbb{P}(Y_k \geq 0) \leq \sum_{k=2}^{K} \mathbb{E}\big[\exp(tY_k/(n_1 n_2 (1-p)))\big] \leq (K-1)e^{-(1-o(1))NI}.$$

Using the assumption $\frac{\log K}{NI} \to 0$, we obtain $\mathbb{P}(\widehat{z}_i(Z^*) \neq z_i^*) \leq e^{-(1-\eta)NI}$. This proves the case $\delta = 0$. Now we compare $Y_k$'s obtained from Algorithm 3 initialized with label matrices $Z^*$ and $Z$, and denoted by $Y_k(Z^*)$ and $Y_k(Z)$, respectively. For all $Z \in B(\delta)$, we show that $|Y_k(Z^*) - Y_k(Z)| \leq 3(n_1 \vee n_k) n\delta$ if $z_i^* = e_1$, giving

$$\mathbb{P}\big(\exists Z \in B(\delta) \text{ such that } \widehat{z}_i(Z) \neq z_i^*\big) \leq \sum_{k=2}^{K} \mathbb{P}\big(Y_k \geq -3(n_1 \vee n_k)n\delta\big).$$

We apply the Chernoff inequality with the same choice of $t$ to obtain (17). We arrive at the proof of Theorem 2. Let $\widehat{Z}(Z)$ be the output of Algorithm 3 with initial label matrix $Z$. Consider the event $\mathcal{A}_\delta = \{\text{Mis}(\widetilde{Z}, Z^*) \leq \delta\}$. For any $\varepsilon > 0$,

$$\mathbb{P}\big(\text{Mis}(\widehat{Z}(\widetilde{Z}), Z^*) > \varepsilon\big) \leq \mathbb{P}\big(\mathcal{A}_\delta^c\big) + \mathbb{P}\big(\exists Z \in B(\delta'), \text{ Mis}(\widehat{Z}(Z), Z^*) > \varepsilon\big).$$

We have $\mathbb{P}\big(\mathcal{A}_\delta^c\big) = o(1)$ under assumption (b1). Letting $\varepsilon = NIe^{-(1-\eta')NI + \frac{3Kn\delta N}{2pn_{\min}}}$, one can verify that the second probability also converges to 0 under the conditions of the theorem and $\varepsilon = e^{-(1-o(1))NI}$. This proves (20) under assumption (b1). For the proof under assumption (b2), please see Appendix D.4

## D.4   DETAILED PROOF OF THEOREM 2

Let $\widehat{Z}(Z)$ be the output of Algorithm 3 with initial label matrix $Z$. Consider the event $\mathcal{A}_{\delta'} = \{\text{Mis}(\widetilde{Z}, Z^*) \leq \delta'\}$. For any $\varepsilon > 0$,

$$\mathbb{P}\big(\text{Mis}(\widehat{Z}(\widetilde{Z}), Z^*) > \varepsilon\big) \leq \mathbb{P}\big(\mathcal{A}_{\delta'}^c\big) + \mathbb{P}\big(\exists Z \in B(\delta'), \text{ Mis}(\widehat{Z}(Z), Z^*) > \varepsilon\big). \tag{18}$$

If assumption (b1) holds, then let $\delta' = \delta$ so that $\mathbb{P}\big(\mathcal{A}_\delta^c\big) = o(1)$. If assumption (b2) holds, Then we let $\delta' = \sqrt{n_{\min}pI\delta/(Kn)}$ so that

$$\frac{Kn\delta'}{n_{\min}pI} = \sqrt{\frac{Kn\delta}{n_{\min}pI}} = o(1)$$

and by Markov's inequality,

$$\mathbb{P}(\text{Mis}(\widetilde{Z}, Z^*) > \delta') \leq \frac{1}{\delta'}\mathbb{E}[\text{Mis}(\widetilde{Z}, Z^*)] \leq \frac{\delta}{\delta'} = \sqrt{\frac{Kn\delta}{n_{\min}pI}} = o(1).$$

Then, (b1) is satisfied with $\delta = \delta'$. Therefore, it is enough to only consider assumption (b1) and let $\delta' = \delta$ for the rest of the proof.

Let $\pi^*$ be the permutation corresponding to $\text{Mis}(\widetilde{Z}, Z^*)$ in assumption (b1), that is, $\pi^* = \text{argmin}_\pi \sum_{i=1}^{n} 1\{\widetilde{z}_i \neq \pi(z_i^*)\}$. Since we can always assume $\pi^*(z^*)$ to be the true label, without loss of generality, we can assume $\pi^* = \text{identity}$. Writing $T_2$ for the second term in (18),

$$T_2 \leq \mathbb{P}\Big(\exists Z \in B(\delta), \sum_{i=1}^{n} 1\{\widehat{z}_i(Z) \neq z_i^*\} > n\varepsilon\Big) \leq \mathbb{P}\Big(\sum_{i=1}^{n} 1\{\exists Z \in B(\delta), \widehat{z}_i(Z) \neq z_i^*\} > n\varepsilon\Big).$$

By Markov's inequality, we obtain

$$T_2 \leq \frac{1}{n\varepsilon}\sum_{i=1}^{n} \mathbb{P}\big(\exists Z \in B(\delta), \widehat{z}_i(Z) \neq z_i^*\big) \leq \frac{1}{\varepsilon}e^{-(1-\eta')NI + \frac{3Kn\delta N}{2pn_{\min}}}. \tag{19}$$

where the second inequality follows from Lemma 2, given assumption (a) of the theorem.

Assumption (a) of the theorem also implies $\frac{3Kn\delta N}{2pn_{\min}} = o(NI)$, so this term can be absorbed into $\eta'NI$, giving $T_2 \leq \frac{1}{\varepsilon}e^{-(1-\eta'')NI}$ for some $\eta'' = o(1)$. Let

$$\varepsilon = NIe^{-(1-\eta'')NI} = e^{-(1-\eta)NI},$$

where $\eta = \eta'' + \frac{\log(NI)}{NI} = o(1)$. It follows from (19) that

$$T_2 \leq \frac{1}{\varepsilon}e^{-(1-\eta'')NI} = \frac{1}{NI} = o(1).$$

Hence, we obtain (12) as desired.

### D.4.1 AN AUXILIARY LEMMA

We state the case $\delta = 0$ in Lemma 2 as a separate lemma and prove it first. Recall that $n_{\min} = \min_{k \in [K]} n_k$ where $n_k$ is the size of the $k$th cluster. We have the following lemma.

**Lemma 3** (Local refinement with $Z^*$)**.** *Suppose the initial label matrix in Algorithm 3 is $Z^*$, and assume $n_{min}p(1 \wedge I)/K \to \infty$ and $\frac{\log K}{NI} \to 0$, then*

$$\mathbb{P}(\widehat{z}_i \neq z_i^*) = e^{-(1-\eta)NI} \tag{20}$$

*for some $\eta = o(1)$. As a direct consequence, $\mathbb{E}[\mathrm{Mis}(\widehat{Z}, Z^*)] \leq e^{-(1-\eta)NI}$.*

*Proof of Lemma 3.* Let $q := p/K$ and $\tilde{p} := (K-1)q := p - q$. We first focus on the probability $\mathbb{P}(\widehat{z}_1 \neq z_1^*)$. Let $\mathcal{C}_k^* = \{i \geq 2 : z_i^* = e_k\}$. We have $b_k = \sum_{i \in \mathcal{C}_k^*} \langle z_i, z_1 \rangle$. Since $z_1^* = e_1$ by assumption, $z_1$ takes values $e_1$ and any of $e_\ell, \ell \neq 1$ w.p. $1 - \tilde{p}$ and $q$. For $i \in \mathcal{C}_1^*$, $z_i$ has the same distribution as $z_1$. For $i \in \mathcal{C}_2^*$, $z_i$ takes values $e_2$ and any of $e_\ell, \ell \neq 2$ w.p. $1 - \tilde{p}$ and $q$ respectively.

Note that $(b_1, b_2)$ is independent of $z_1$. It follows that

$$(b_1, b_2) \mid z_1 \sim \begin{cases} \mathrm{Bin}(n_1, 1 - \tilde{p}) \otimes \mathrm{Bin}(n_2, q), & \text{if } z_1 = e_1 \\ \mathrm{Bin}(n_1, q) \otimes \mathrm{Bin}(n_2, 1 - \tilde{p}), & \text{if } z_1 = e_2 \\ \mathrm{Bin}(n_1, q) \otimes \mathrm{Bin}(n_2, q), & \text{if } z_1 \notin \{e_1, e_2\} \end{cases} \tag{21}$$

where $\otimes$ is the notation for the product measure, that is, $b_1$ and $b_2$ are independent in each case. The three possibilities above hold with probability $1 - \tilde{p}$, $q$ and $(K-2)q$ respectively. Let $Y = n_1 b_2 - n_2 b_1$ and let $M_Y(\lambda)$ be the moment-generating function (MGF) of $Y$.

Let $\psi(\lambda; p) = 1 - p + pe^\lambda$ be the MGF of a $\mathrm{Ber}(p)$ variable. Then, the MGF of $\mathrm{Bin}(n, p)$ is $\psi(\lambda; p)^n$ and hence

$$\mathbb{E}[e^{\lambda Y} \mid z_1] = \mathbb{E}[e^{\lambda n_1 b_2} \mid z_1] \cdot \mathbb{E}e^{-\lambda n_2 b_1} \mid z_1]$$

$$= \begin{cases} \psi(\lambda n_1; q)^{n_2} \cdot \psi(-\lambda n_2; 1 - \tilde{p})^{n_1} & \text{if } z_1 = e_1 \\ \psi(\lambda n_1; 1 - \tilde{p})^{n_2} \cdot \psi(-\lambda n_2; q)^{n_1} & \text{if } z_1 = e_2 \\ \psi(\lambda n_1; q)^{n_2} \cdot \psi(-\lambda n_2; q)^{n_1} & \text{if } z_1 \notin \{e_1, e_2\}. \end{cases}$$

Let $\phi(\lambda; \mu) = \exp(\mu(e^\lambda - 1))$ be the MGF of $\mathrm{Poi}(\mu)$ and note that $\psi(\lambda; p)^n \leq \phi(\lambda; np)$. Then, for example, we have

$$\mathbb{E}[e^{\lambda Y} \mid z_1 = e_1] \leq \phi(\lambda n_1; n_2 q) \cdot \phi(-\lambda n_2; n_1(1 - \tilde{p})).$$

Since $\phi(\lambda; \mu) = \exp[\mu(\lambda + o(\lambda))] = \exp[\mu\lambda(1 + o(1))]$ for $\lambda = o(1)$, we obtain

$$\mathbb{E}[e^{\lambda Y} \mid z_1 = e_1] \leq \exp[n_1 n_2 \lambda q(1 + o(1)) - n_1 n_2 \lambda(1 - \tilde{p})(1 + o(1))]$$

assuming that $\lambda(n_1 + n_2) = o(1)$. Then,

$$\mathbb{E}[e^{\lambda Y} \mid z_1 = e_1] \leq \exp[n_1 n_2 \lambda(q - 1 + \tilde{p})(1 + o(1))]$$

Take $\lambda = t[n_1 n_2(1 - p)]^{-1}$ for some $t \geq 0$ to be determined below. Noting $q - 1 + \tilde{p} = -(1 - p)$,

$$\mathbb{E}[e^{\lambda Y} \mid z_1 = e_1] \leq \exp[-t(1 + o(1))].$$

The case $z_1 = e_2$ is argued similarly and we obtain the bound $\mathbb{E}[e^{\lambda Y} \mid z_1 = e_2] \leq \exp[t(1+o(1))]$. For $z_1 \notin \{e_1, e_2\}$, we perform a second-order expansion, assuming $\lambda = o(1)$:

$$\phi(\lambda; \mu) = \exp\big[\mu\big(\lambda + \tfrac{1}{2}\lambda^2 + o(\lambda^2)\big)\big] \leq \exp\big[\mu\big(\lambda + \lambda^2\big)\big]$$

and obtain

$$\psi(\lambda n_1; q)^{n_2} \cdot \psi(-\lambda n_2; q)^{n_1} \leq \exp\big[\lambda^2 n_1 n_2 (n_1 + n_2) q\big].$$

Let $\gamma := 2q/(1-p)^2$ and let $n_{\mathrm{har}} := 2n_1 n_2/(n_1 + n_2)$ be the harmonic mean of $n_1$ and $n_2$. Note that $n_{\mathrm{har}} \geq n_{\min}$. We have

$$\lambda^2 n_1 n_2 (n_1 + n_2) q = \frac{t^2 (n_1 + n_2) q}{n_1 n_2 (1-p)^2} = \gamma t^2 / n_{\mathrm{har}}.$$

To summarize, the conditional MGF satisfies

$$\mathbb{E}[e^{tY/(n_1 n_2(1-p))} \mid z_1] \leq \begin{cases} \exp\big[-t\big(1 + o(1)\big)\big] & \text{if } z_1 = e_1 \\ \exp\big[t\big(1 + o(1)\big)\big] & \text{if } z_1 = e_2 \\ \exp(\gamma t^2 / n_{\mathrm{har}}) & \text{if } z_1 \notin \{e_1, e_2\}. \end{cases}$$

Recall that the events $z_1 = e_1$, $z_1 = e_2$ and $z_1 \notin \{e_1, e_2\}$ happen with probability $1 - \tilde{p}$, $q$ and $(K-2)q$ respectively. It follows that

$$M_Y(t/(n_1 n_2(1-p))) \leq (1 - \tilde{p}) e^{-t(1+o(1))} + q e^{t(1+o(1))} + (K-2)q e^{\gamma t^2/n_{\mathrm{har}}}. \qquad (22)$$

Let us set

$$t = \frac{1}{2} \log((1-\tilde{p})/q) = \frac{1}{2} \log\Big(1 + \frac{K}{p}(1-p)\Big), \qquad (23)$$

so that $(1-\tilde{p})e^{-t} = qe^t = \sqrt{(1-\tilde{p})q}$. Then, $t \geq 0$ and since $\log(1+x) \leq x$, we have

$$t \leq K(1-p)/(2p). \qquad (24)$$

The condition $\lambda(n_1 + n_2) = o(1)$ is satisfied under assumption $n_{\min} p / K \to \infty$, since

$$\lambda(n_1 + n_2) = \frac{(n_1 + n_2)t}{n_1 n_2 (1-p)} = \frac{2t}{n_{\mathrm{har}}(1-p)} \leq \frac{K(1-p)/p}{n_{\mathrm{har}}(1-p)} \leq \frac{K}{n_{\min} p} = o(1).$$

Recalling that $q = p/K$, the exponent of the last term in (22) satisfies

$$\frac{\gamma t^2}{n_{\mathrm{har}}} \leq \frac{\gamma K^2 (1-p)^2}{4p^2 n_{\mathrm{har}}} = \frac{2qK^2}{4p^2 n_{\mathrm{har}}} = \frac{K}{2n_{\mathrm{har}} p} \leq \frac{K}{2n_{\min} p} = o(I)$$

under the assumption of the lemma. It follows that

$$M_Y(t/(n_1 n_2(1-p))) \leq 2\sqrt{(1-\tilde{p})q}\, e^{o(t)} + (K-2)q e^{o(I)}$$
$$= 2(\sqrt{(1-\tilde{p})q})^{1+o(1)} + (K-2)q e^{o(I)}$$
$$= \big[(\sqrt{(1-\tilde{p})q})^{o(1)} \vee e^{o(I)}\big]\, e^{-I}$$

where the first equality is by $e^{o(t)} = (e^t)^{o(1)} = (\sqrt{(1-\tilde{p})q})^{o(1)}$ for our choice of $t$, and the second equality by the definition (11) of $I$. Since $\sqrt{(1-\tilde{p})q} \leq e^{-I}$, we have $(\sqrt{(1-\tilde{p})q})^{o(1)} = e^{o(I)}$, hence

$$M_Y(t/(n_1 n_2(1-p))) \leq e^{o(I)} e^{-I} = e^{-(1-o(1))I} = e^{-(1-\eta)I}.$$

Let $Y_1, \ldots, Y_N$ be the i.i.d. copies of $Y$. By Markov's inequality,

$$\mathbb{P}\Big(\sum_{j=1}^N Y_j \geq 0\Big) = \mathbb{P}\Big(e^{\lambda \sum_{j=1}^N Y_j} \geq 1\Big) \leq \mathbb{E}e^{\lambda \sum_{j=1}^N Y_j} = M_{Y_1}(\lambda)^N \leq e^{-(1-\eta)NI}.$$

The above argument shows that $\mathbb{P}\big(\frac{b_2}{n_2} \geq \frac{b_1}{n_1}\big) \leq e^{-(1-\eta)NI}$. Repeating the argument for the $i$th label, it shows that

$$\mathbb{P}(\widehat{z}_i(Z^*) \neq z_i^*) \leq \mathbb{P}\Big(\max_{k=2,\ldots,K} \frac{b_k}{n_k} \geq \frac{b_1}{n_1}\Big) \leq \sum_{k=2}^K \mathbb{P}\Big(\frac{b_k}{n_k} \geq \frac{b_1}{n_1}\Big) \leq (K-1)e^{-(1-\eta)NI}.$$

If $K = 2$, then we have already obtained (20). If $K > 2$, then

$$(K-1)e^{-(1-\eta)NI} = e^{-(1-\eta)NI + \log(K-1)} = e^{-(1-\eta)NI + o(NI)}.$$

The term $o(NI)$ can be absorbed into $\eta NI$, so we can still obtain (20). $\qquad \square$

### D.4.2 DETAILED PROOF OF LEMMA 2

Let $i = 1$ without loss of generality, and let $Z \in B(\delta)$, and $\hat{n}_k = n_k(Z)$, the size of the $k$th cluster in the label matrix $Z$. Let $b_k(Z^*) = (Z^*_{-1} \bar{X}_1)_k$ and $b_k(Z) = (Z_{-1} \bar{X}_1)_k$ where $\bar{X}_1$ is the first column of $\bar{X}$ in the algorithm. Suppose $Z$ has at most $n\delta$ labels different from $Z^*$, then

$$\left| [n_1 b_2(Z^*) - n_2 b_1(Z^*)] - [n_1 b_2(Z) - n_2 b_1(Z)] \right| \leq (n_1 \vee n_2) n\delta.$$

and

$$\left| [n_1 b_2(Z) - n_2 b_1(Z)] - [\hat{n}_1 b_2(Z) - \hat{n}_2 b_1(Z)] \right| \leq |n_1 - \hat{n}_1| \cdot b_2(Z) + |n_2 - \hat{n}_2| \cdot b_1(Z)$$
$$\leq n\delta (n_1 + n_2).$$

Let $Y(Z^*) = n_1 b_2(Z^*) - n_2 b_1(Z^*)$ and $Y(Z) = \hat{n}_1 b_2(Z) - \hat{n}_2 b_1(Z)$. Combining the two by the triangle inequality, we obtain

$$|Y(Z^*) - Y(Z)| \leq 3(n_1 \vee n_2) n\delta =: h(\delta).$$

By Markov's inequality, for $\lambda \geq 0$,

$$\mathbb{P}\left( \max_{Z \in B(\delta)} Y(Z) \geq 0 \right) \leq \mathbb{P}\left( Y(Z^*) \geq -h(\delta) \right)$$
$$= \mathbb{P}\left( e^{\lambda N Y(Z^*)} \geq e^{-\lambda N h(\delta)} \right) \leq e^{\lambda N h(\delta)} \mathbb{E}[e^{\lambda N Y(Z^*)}]$$

As in the proof of Lemma 3, we take $\lambda = t [n_1 n_2 (1-p)]^{-1}$, with $t$ given by (23). Using the upper bound (24), we have

$$\lambda N h(\delta) = \frac{3(n_1 \vee n_2) n N}{n_1 n_2 (1-p)} \delta t \leq \frac{3 K n \delta N}{2 p n_{\min}}.$$

The result follows as in Lemma 3.

## E   RPM AND BAYESIAN AGGREGATION

One might ask whether RPM is a useful model in practice. For the applications in which all the label vectors are perturbations of a common true "center", and our goal is to recover this center, RPM is a good first approximation. This is the case for Bayesian label aggregation as we argue below. In such settings, the RPM is like the i.i.d. noise model used in classical regression. Although one can imagine more complex regression models (like those with heteroscedastic noise, or mixtures of regressions, etc.), the i.i.d. setting still provides a lot of insights for understanding the more complex models.

### E.1   RPM IS A GOOD MODEL FOR A CONCENTRATED POSTERIOR

Lets us now argue how one can arrive at RPM in the context of Bayesian aggregation, by systematically making some assumptions. First, we note that our goal in the paper is not to prove the "posterior concentration around the truth", also known as *posterior consistency*. This is problem-specific and out of the scope of this work. We assume that we are working with a model for which posterior consistency has already been established. The question that we are trying to answer is then this:

> Given that the posterior concentrates around the true partition, and given that the MCMC has converged—that is, we are sampling from this concentrated posterior—can we obtain a consistent estimate of the center of the posterior (which would be the true partition) based these samples?

For this purpose, it is enough to assume that we are observing samples from the posterior $p(z_1, \ldots, z_n | D)$, where $D$ is the observed data, and this posterior is concentrating around $z^*$ which is the true partition. For simplicity, let us drop $D$ and note that the posterior is some multivariate discrete distribution $p(z_1, \ldots, z_n)$. Then, we proceed in steps:

1. First, we address the label-switching issue. Let $z$ be a sample from the posterior and let permutation $\pi$ be the minimizer of $H(z^*, \pi(z))$ over all $K!$ permutations, where $H(\cdot, \cdot)$ is the Hamming distance. We note that $\pi(z)$ is an equivalent label vector to $z$ (only the cluster

labels are permuted.) We consider the distribution of $\pi(z)$ as the posterior distribution rather than that of $z$. This is only to simplify the discussion and is without loss of generality, since our methods are invariant to label-switching. The distribution of $\pi(z)$ will have a single mode at $z^*$ while that of $z$ will have $K!$ modes on all equivalent versions of $z^*$. Given such simplification, renaming $\pi(z)$ to $z$ from now on, the posterior is a multivariate distribution $p(z_1, \ldots, z_n)$ which is highly concentrated around $z^* = (z_1^*, \ldots, z_n^*)$. This follows from the posterior consistency assumption.

2. We claim that this multivariate distribution can be approximated by the product of its marginals $p(z_1, \ldots, z_n) \approx \prod_{i=1}^{n} p_i(z_i)$. This is intuitively clear for any concentrated discrete distribution. Alternatively, it is intuitively clear to anyone who has looked at MCMC samples at stationarity. Each node/object $i$ usually has a most likely assignment which is $z_i^*$, but it occasionally jumps to some other label with a small probability. The fluctuations for different nodes are independent; this is intuitively because the bulk of the labels don't change their clusters all at once; only a few do at any given time.

3. *Non-uniform RPM*: Given the independence assumption across $i$, the most general form $p_i(z_i)$ can take is a categorical (a.k.a. Multinomial$(1,\pi)$) distribution. The bulk of the mass of this categorical variable will be on $z_i^*$, and the rest distributed among the other labels. For example, for some node, $i$, the label can jump, say, between $z_i^* = 3$ and 5 with the bulk of the probability on 3. For others it could be that when they jump from $z_i^*$, they land over a larger collection of labels. Here, we are making the simplifying assumption that for each node, the mass that is put on anything other than $z_i^*$ is uniformly distributed over the label set $[K] \setminus \{z_i^*\}$. This assumption can be removed, and we can work with general categorical variables, at the expense of making the rates and the analysis more complicated.

4. *Inhomogeneous RPM*: Given that the noise in the categorical variable is uniform over $[K] \setminus \{z_i^*\}$, we now assume that the probability of taking any of those values is the same for the all nodes (i.e., independent of $i$). This is exactly the homogeneous RPM that we consider. This assumption is easy to remove and we can work with the inhomogeneous RPM that allows this probability to depend on $i$.

We do not lose anything in Step 1. Steps 3 and 4 are simplifying steps that are taken for the ease of understanding and presentation. The main assumption is Step 2, the approximation by the product of marginals. Below we provide some hard evidence that this is very reasonable in the Bayesian setting.

### E.2 Hard Evidence of Near-Independence

We consider the problem of recovering the clusters in a stochastic block model (SBM) which is a random network model with latent node clusters. Figure 5 shows the Sequential NMI plot for a Gibbs sampler on a non-parametric SBM (with a Dirichlet Process prior on labels). The sequential NMI means that we compute the NMI of the partition at iteration $t$ relative to that at iteration $t - 1$, and the x-axis shows $t$. The plot suggests that the chain enters stationarity roughly around iteration 100.

We use the MMD-based approach, described in (19; 20; 38), to compare the posterior joint distribution of the labels to its approximate versions:

- Figure 6(a) shows the result for samples from iterations 100 to 1000 of the Gibbs sampler. This is the stationary joint distribution.

- Figure 6(b) shows the result for samples from iterations 1 to 100 of the Gibbs sampler. This is based on the transient samples (effectively, average joint behavior over the transient period).

- Figure 6(c) shows the results on a movie rating dataset with complicated dependent joint distribution (that has nothing to do with SBM).

We refer to (19; 20; 38) for details of how these experiments are performed. The bootstrap MMD serves as the baseline; those methods whose MMD is closer to the bootstrap are better approximations of the joint distribution. In general smaller MMD means a better approximation of the joint. Ind Mult is exactly the approximation by independent multinomials (i.e. product of marginals). The Copula Mult is a good joint model for the discrete multivariate distribution.

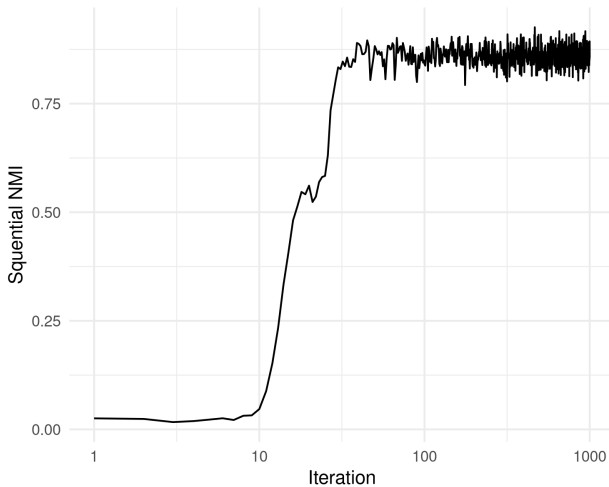

Figure 5: Sequential NMI for the SBM Gibbs sampler

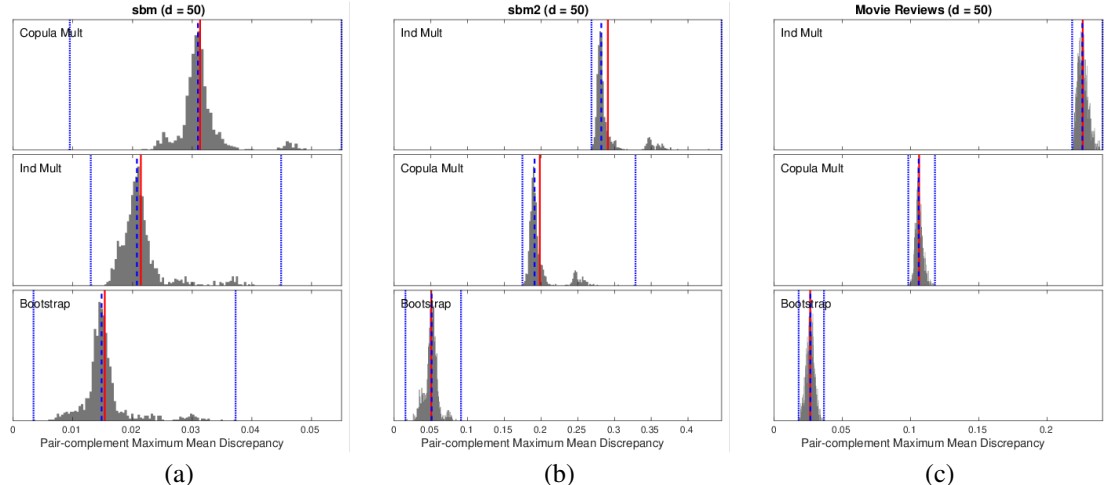

Figure 6: MMD histograms between the posterior distribution and its various approximations

We see that for the movie rating data and the transient chain, indeed Copula Mult has a much lower MMD, than Ind Mult, showing that there is dependence in the joint that is not captured by Ind Mult. However, for the stationary distribution (Figure 6(a)), the Ind Mult has comparable (and even slightly smaller MMD) relative to the Copula Mult, and is close to bootstrap. This shows that the product of marginals is a good approximation in this case, and justifies Step 2 in our reduction.

## F    COMPARISON WITH STOCHASTIC BLOCK MODEL

Below we outline some of the similarities and differences between the problem of cluster recovery in SBM and the consensus clustering problem we consider in this paper. Suppose that $A$ is the adjacency matrix generated from the stochastic block model (SBM) and $X$ is the association matrix generated from the RPM.

**Similarity:**    A larger value of $X_{ij}$ increases the likelihood of $i$ and $j$ being in the same cluster in RPM. Similarly, $A_{ij} = 1$, i.e., there is an edge between $i$ and $j$, increases the likelihood of $i$ and $j$ being in the same community in SBM. Therefore, both $X$ and $A$ can be considered proximity matrices and we can utilize a min-cut algorithm on them to find the clusters. To approximate the

min-cut algorithm, various researchers have proposed the approach of using a good initialization plus a local refinement step. This idea can be applied to many clustering problems, including community detection. In this paper, we show that it can also be applied to consensus clustering.

**Difference:** The entries of the adjacency matrix $A$ are independent, but the entries of the association matrix $X$ are not. Indeed, the entries on the same row of $X$ have very strong dependence. The likelihood function of $X$ is very different from $A$, but we can still show that a simple local refinement step outputs optimal labels. The error rate is comparable to the Bayes error rate given by likelihood ratio test.

