# OpenReview forum: "Statistical Guarantees for Consensus Clustering"
_ICLR.cc/2023/Conference — ICLR 2023 poster_

### Official Review · Reviewer_QD6L · 2022-10-23

**Confidence:** 2
**Correctness:** 4
**Technical Novelty And Significance:** 3
**Empirical Novelty And Significance:** Not applicable
**Recommendation:** 8

**Clarity, Quality, Novelty And Reproducibility:**

The paper is very well structured and written, with the important technical details proved in the supplementary. A few clarifications for some minor details: (1) In definition (1), the permutation matrix $P$ is sampled from all permutation matrices, right? In particular, if $Z_{1}$ and $Z_{2}$ are sampled from the model, the corresponding $P_{1}$ and $P_{2}$ can be different if I understand correctly. (2) In most cases, the refinement procedure improves the solution quality. But for the green curve (CC Pivot), the result is the reverse, is there any explanation?

**Strength And Weaknesses:**

The statistical consistency under the random perturbation model has been proved for the proposed algorithms. In particular, the proposed local refinement procedure is interesting, which is motivated by matching the error rate in the supervised consensus clustering. This is an important technical contribution. There are a few aspects I believe the authors can further elaborate: (1) RPM has a strong connection with the Bayesian aggregation, as shown in the supplementary. Have other probabilistic models been considered in literature? Empirically, how robust is the proposed algorithm against other settings? (2) Computationally, by lifting the solution space to the association matrices, the dimension is $n\times n$, would it be too expensive when $n$ is large?

**Summary Of The Paper:**

This work studies the unsupervised consensus clustering problem: given $N$ $K$- clustering vectors on n data points, a single clustering solution is to be found that minimizes the distance to $N$ $K$- clustering vectors. This problem is in general challenging due to the freedom of label switching among $N$ $K$ - clustering vectors. The authors address the label-switching issue by lifting the solution space to association matrices, and relaxing the original problem to a semi-definite programming (SDP) problem that has a closed-form solution. Afterwards, the clustering solution can be constructed from the SDP solution. The proposed algorithm achieves statistical consistency, i.e., the error bound goes to 0 as $n,N\to\infty$, under the random perturbation generative model of these $N$ $K$- clustering vectors. Moreover, a further refinement procedure on the clustering solution can be shown to attain the near optimal error rate with respect to $N$. The authors corroborate the theory by evaluating the proposed algorithm under different choice of n, N and data balancedness. A quite comprehensive comparison with other algorithms in the literature has also been done in this work.

**Summary Of The Review:**

This work makes a contribution for establishing the statistical consistency and near optimality rate for the unsupervised consensus clustering problem, under a generative model. The theory is interesting to the community.

---

> ### Author Response · Authors · 2022-11-14
> **Response to comments**
>
> Thanks for your positive feedback and assessment.
>
> >RPM has a strong connection with the Bayesian aggregation, as shown in the supplementary. Have other probabilistic models been considered in literature? Empirically, how robust is the proposed algorithm against other settings?
>
> As far as we have searched, the literature on consensus clustering (and related areas)  is often not concerned with probabilistic models of the problem. An exception is perhaps the work of Goder and Filkov [18] where a model similar to RPM is considered for empirical evaluation. But, we are not aware of any work that does a theoretical analysis under RPM, nor any work that proposes an alternative probabilistic model for that matter. We agree that it would be interesting to see empirical performance against other settings, but at the moment we can no think of a model substantially  different from RPM to test against.
>
> >Computationally, by lifting the solution space to the association matrices, the dimension is $n\times n$, would it be too expensive when $n$ is large?
>
> Thanks for pointing this out. Yes, lifting generally increases computational complexity.
> Considering Algorithm 1 following by Algorithm 3, the most expensive step is to find the $K$-means of $X$ in Algorithm 1. A $(1+\epsilon)$-approximation of $K$-means runs in $O(n^2)$ time. If we apply the spectral clustering algorithm in Algorithm 2, the run time of $K$-means can be reduced to $O(nK)$, but eigenvalue decomposition (EVD) takes $O(n^3)$ time in the worst-case. In practice, it could run faster on generic problems,  especially since we care about the top $K$ eigenvalues.
>
> It is worth noting that $O(n^2)$ or $O(n^3)$ is relatively cheap. In the applications where the label vectors are generated from the posterior distribution or Gibbs sampler, the algorithm generating these label vectors usually takes a longer time to run.
>
> >In definition (1), the permutation matrix $P$ is sampled from all permutation matrices, right? In particular, if $P\_1$ and $P\_2$ are sampled from the model, the corresponding $Z\_1$ and $Z\_2$ can be different if I understand correctly.
>
> Yes, you are correct. Even if $Z\_1'=Z\_2'$ (the label matrix before permutation) $Z\_1$ and $Z\_2$, can be different because of their different permutations.
>
> >In most cases, the refinement procedure improves the solution quality. But for the green curve (CC Pivot), the result is the reverse, is there any explanation?
>
> It is an interesting observation (although not always the case). We do not have an explanation for it at the moment.

---

### Official Review · Reviewer_38hB · 2022-10-25

**Confidence:** 4
**Correctness:** 3
**Technical Novelty And Significance:** 2
**Empirical Novelty And Significance:** 2
**Recommendation:** 5

**Clarity, Quality, Novelty And Reproducibility:**

- Are the relaxations tight? See above.
- What is the relationship between RPM and SBM, in terms of the association matrix? Can one be subsumed by the other?
- It seems to me that Eq. (5) does not parse, since both Z and P's are variables in the argmin, but the left hand side is only Z. Should it be \argmin_Z \min_{P_1,\dots,P_N}?
- Page 3: "One can verify that problem (5) is equivalent to (3)." Could you give a brief proof?

**Strength And Weaknesses:**

Strength:
- The paper is easy to follow in general. Most of the notations are properly defined.
- The flow of the analysis is clear (original problem -> matrix problem -> SDP -> algorithms).

Weakness:

My biggest concern is that the problem being studied feels artificial, and a lot of techniques used are standard and very similar to existing community detection / stochatic blockmodel (SBM) literature.

About the problem:
- I feel that the problem is intentionally posed in the current way to make the analysis more complicated than necessary. For example, the authors stress on Page 2, that problem (3) is hard because of the factorial number of permutations. This is exaggerating because one can simply consider the association matrices zz^T, and this is a quite common step in community detection literature because very often people do not care about permutations.
- If the authors believe that the step by step relaxation is necessary (instead of going for xx^T's directly), it would be helpful if they show the tightness of their relaxation. In other words, is there any proof governing that, under what models or statistical conditions, the optimal solution to (7) is equal to the optimal solution to (6), and the optimal solution to (5)? Otherwise, such relaxation is heuristic and without guarantee.
- How should I interpret RPM? Why is it reasonable? Is there any real world dataset follows from the model?

About novelty:
- The vector -> matrix -> SDP relaxation is standard in SBM works; for instance see SDP-1 in [1].
- The idea of using a permutation matrix is covered in a highly similar fashion between (3.3) and (3.4) in [1].
- In fact, I fear that the problem of minimizing the Frobenious norm in this paper is equivalent to the problem of maximizing the inner product as in SDP-1 [1], by adding some entrywise summation constraints like X \1 = (n/k) \1.

References:
- [1] Amini, Arash A., and Elizaveta Levina. "On semidefinite relaxations for the block model." The Annals of Statistics 46.1 (2018): 149-179.

**Summary Of The Paper:**

The paper studies the problem of computing consensus clustering when groups can be equivalently permuted. They analyzed the problem in the discrete vector form, combinatorial matrix form, and nonnegative semidefinite programming (SDP) form by relaxation, and propose a simple weighting algorithm to compute the low rank matrix and consequently the labeling. To provide some guarantees for their proposed algorithm, the authors propose a random perturbation model (RPM) which generates the aforementioned labeling instances in a stochastic way. Under this RPM setting, the authors provide some upper bounds with respect to the misclassification rate.

**Summary Of The Review:**

The problem setting and analysis feel artificial, and the techniques are standard.

---

> ### Author Response · Authors · 2022-11-14
> **Response to comments**
>
> Thanks for your positive feedback. Please see below for our responses to your concerns. We break it into two parts due to the space constraint.
>
> > My biggest concern is that the problem being studied feels artificial, and a lot of techniques used are standard and very similar to existing community detection and stochatic blockmodel (SBM) literature.
>
> The problem is not artificial. This exact problem is studied under many different names including "consensus clustering" in various papers from a diverse array of fields as we point out in the related work. We will elaborate more below. There is indeed a close connection to community detection in SBM, but there is also notable differences which makes a theoretical analysis of the problem challenging.
>
> > I feel that the problem is intentionally posed in the current way to make the analysis more complicated than necessary. For example, the authors stress on Page 2, that problem (3) is hard because of the factorial number of permutations. This is exaggerating because one can simply consider the association matrices $zz\^T$, and this is a quite common step in community detection literature because very often people do not care about permutations.
>
> It is not intentionally posed that way; this is how we receive the data in consensus clustering (and  Bayesian aggregation). It is true that one can get rid of the permutation by passing to association matrices, but the key is this: Doing so will introduce **dependence among the entries** of these association matrices. Please note that it is not just the variable $Z$ that needs to be passed to an association matrix $ZZ\^T$, but each input label matrix $Z\_i$ needs to be transformed to $X\_i = Z\_i Z\_i\^T$. It is reasonable to assume that the perturbations over rows of $Z\_i$ are independent, but the resulting matrix $X\_i$ will have all its entries highly dependent. This is the challenging aspect of the problem which sets it apart from the SBM setting. Despite close similarities, in the SBM setting, the entries of the observed adjacency matrix are independent given the latent label. Here the aggregated association matrix has highly dependent entries.
>
> Also, as noted by reviewer QD6L, lifting to association matrices **increases computational complexity**. It would be cheaper if we can work with the original clusterings and deal with the permutation ambiguity in a different way. Although, at the moment we don't know of any algorithm that can do so with provable guarantees.
>
> In short, the transformation is simple. Its theoretical consequences are not.
>
> > If the authors believe that the step by step relaxation is necessary (instead of going for $xx^T$'s directly), it would be helpful if they show the tightness of their relaxation ...
>
> Our discussion of the relaxation is to show how these problems are connected and to argue that the spectral clustering approach we propose is closely related to the Mirkhin barycenter problem. There is not much incentive to show the tightness of the relaxation, since the original problem (the Mirkhin or Miss barycenter) are not necessarily optimal themselves (or it is unknown if they are). This is in contrast to the SBM literature where the SDPs are relaxations of the maximum likelihood estimator which is known to be optimal in some regimes.
>
> In other words, even if we show tightness of the relaxation, there is no reason that the original barycenter problems have solutions that are statistically consistent or optimal. Instead, we use this part of the paper as a motivation of the method and show the consistency and optimality of the method directly.
>
> > How should I interpret RPM? Why is it reasonable? Is there any real world dataset follows from the model?
>
> We have an extensive discussion of why and when RPM is a reasonable model in Appendix E of the paper (RPM and Bayesian Aggregation). In short, RPM is a very reasonable first model, useful in cases where one believes the various input clusterings are perturbation of a single central clustering. This, for example, is very reasonable in the Bayesian label aggregation problem, where one tries to find the center of the posterior, as discussed in Appendix E.1 and E.2.
>
> Please see the next posted comment for the rest ...

---

> > ### Author Response · Authors · 2022-11-14
> > **Connection to SBM and SDP-1**
> >
> > > About novelty: The vector -> matrix -> SDP relaxation is standard in SBM works; for instance see SDP-1 in [1].
> > The idea of using a permutation matrix is covered in a highly similar fashion between (3.3) and (3.4) in [1].
> >
> > Thanks for pointing out to [1]. We will cite that work in the paper. Our approach in deriving the SDP is very similar to [1]. However, please note that the SDP relaxation, etc. is not the main contribution of our work. The subsequent theoretical analysis is our main contribution which is different from the SBM setting.
> >
> > > In fact, I fear that the problem of minimizing the Frobenious norm in this paper is equivalent to the problem of maximizing the inner product as in SDP-1 [1], by adding some entrywise summation constraints like $X 1 = (n/k) 1$.
> >
> > It is close, but it is not exactly that since $\\| X\\|\_F^2$ in our case is allowed to vary. We are also not enforcing balancedness of the clusters which the constraint $ X 1 = (n/k) 1$  entail. In any case, the SDP relaxation in our work is just a motivating point for the estimator that we analyze. The main contribution is the analysis and the proof of the optimality of the proposed estimator. Even if our estimator was exactly SDP-1 (which is not), analyzing it in the consensus clustering setting would be nontrivial due the dependence of the entries of $\bar X$ as we pointed out earlier.
> >
> > >Are the relaxations tight? See above.
> >
> > Please see our response there. In short, although an interesting question, it doesn't matter for this problem. We analyze the relaxation directly.
> >
> > > What is the relationship between RPM and SBM, in terms of the association matrix? Can one be subsumed by the other?
> >
> > Suppose $A$ is the adjacency matrix generated from SBM and $X$ is the association matrix generated from RPM.
> >
> > **Similarity**: A larger value of $X\_{ij}$ increases the likelihood of $i$ and $j$ being in the same cluster in RPM. Similarly, $A\_{ij}=1$, i.e., there is an edge between $i$ and $j$, increases the likelihood of $i$ and $j$ being in the same community in SBM. Therefore, both $X$ and $A$ can be considered proximity matrices and we can utilize a min-cut algorithm on them to find the clusters. To approximate the min-cut algorithm, various researchers have proposed the approach of using a good initialization plus a local refinement step. This idea can be applied to many clustering problems, including community detection. In this paper, we show that it can also be applied to consensus clustering.
> >
> > **Difference**: The entries of the adjacency matrix $A$ are independent, but the entries of the association matrix $X$ are not. Indeed, the entries on the same row of $X$ have very strong dependence. The likelihood function of $X$ is very different from $A$, but we can still show that a simple local refinement step outputs optimal labels. The error rate is comparable to the Bayes error given by likelihood ratio test.
> >
> > > It seems to me that Eq. (5) does not parse, since both Z and P's are variables in the argmin, ....
> >
> > Yes. Thanks for pointing this out. We will fix it in the revision.
> >
> > > Page 3: "One can verify that problem (5) is equivalent to (3)." Could you give a brief proof?
> >
> > Problem (5) is the matrix version of problem (3).  First, note that in our notation, the $i$th column of $Z$ or $Z\_j$ represents the label of the $i$th object. Let $P\_j$ be the permutation matrix of $\pi\_j$. Then, the $i$th column of $P\_j Z\_j$ corresponds to the permuted label $\pi\_j(z\_{ji})$. If $z\_i\ne \pi\_j(z\_{ji})$, then the $i$th column of $Z-P\_jZ\_j$ has a $+1$ and a $-1$, otherwise all the entries will be zero. The square of the $\ell\^2$-norm of this column is then either 2 or 0.  This shows that $1\\{z\_i\ne \pi\_j(z\_{ji})\\} = \frac 12 [(Z-P\_j Z\_j)\_{*i}]\^2$. Summing over $i$ gives the Frobenius norm on the right-hand side and the rest follows.

---

### Official Review · Reviewer_12P5 · 2022-10-25

**Confidence:** 4
**Correctness:** 4
**Technical Novelty And Significance:** 2
**Empirical Novelty And Significance:** Not applicable
**Recommendation:** 6

**Clarity, Quality, Novelty And Reproducibility:**

The paper is well written overall. The algorithm is along well-studied lines, but the analytical model is new, to the best of my knowledge.

The work is primarily theoretical, so the reproducibility question is not applicable.

**Strength And Weaknesses:**

The main strength of the paper is in analyzing the consistency, especially in terms of the number of misclassifications. They also do so for a natural algorithm.

That said, the results are not very surprising given the generative model. Also algorithmically, it seems that other than improving the N dependence, there was not sufficient novelty, especially since all the results are theoretical. So while I am overall positive about the paper, I don't see it as a definite accept.

Here are some questions that I would like to see the authors address:

1. While the dependence on N is analyzed, the dependence on the other parameters such as \beta is not. Can the authors say if they expect the dependence to be improvable?

2. The kinds of issues the authors run into in terms of cluster recovery are similar to those in the following recent paper: https://proceedings.mlr.press/v178/gamlath22a.html

This is a recent paper so if not a detailed response, it would be good to see the authors' initial thoughts.

**Summary Of The Paper:**

The paper studies the problem of aggregating clusterings of a set of data points. This is a well-studied problem in we are given a collection of clusterings of a given dataset and the goal is to produce a new clustering that is "close" to the given ones.

The problem is non-trivial because clusterings remain the same even after permuting the labels. Thus the authors produce a natural algorithm based on looking at the association (or similarity) matrices of the clusterings and taking an average of those. This is a natural idea, and it has been extensively studied in the literature as the authors also note.

The main contribution of the paper is to provide (a) a consistency analysis of such a procedure under a generative model for clusterings, and (b) giving an algorithm that can achieve an error that drops optimally with the number of clusterings. The generative model is what the authors call the "robust perturbation model", and is a natural one in this context: a set of clusterings are obtained from a "ground truth" clustering by permuting the labels and adding IID error to them. Given these clusterings, the goal is to recover the ground truth up to a permutation.

The paper shows that the simple algorithm that first averages the association matrices and then performs a K-means step achieves a consistency guarantee that goes to zero with the number of clusterings 'N' and the number of points 'n'. They then give an algorithm that improves the dependence on N via a post-processing step.

**Summary Of The Review:**

(Pasted from above): The main strength of the paper is in analyzing the consistency, especially in terms of the number of misclassifications. They also do so for a natural algorithm. That said, the results are not too surprising given the generative model. Also algorithmically, it seems that other than improving the N dependence, there was not sufficient novelty, especially for a theory paper. So while I am overall positive about the paper, I don't see it as a definite accept.

---

> ### Author Response · Authors · 2022-11-14
> **Response to comments**
>
> Thanks for your positive feedback.
>
> > That said, the results are not very surprising given the generative model.
>
> We respectfully disagree. There are surprises. We show that given the generative model, a very popular algorithm (BestOFK) is in general inconsistent, despite being a 2-factor approximation to the median partition problem. This shows that consistency in this problem is not to be taken for granted. That we can even come close to optimality with a simple algorithm is another surprise.
>
> (It is worth noting that BestOFK is proposed in  different fields without noticing that it is the same algorithm; see the 2nd paragraph of Section 2.3.)
>
> >While the dependence on $N$ is analyzed, the dependence on the other parameters such as $\beta$ is not. Can the authors say if they expect the dependence to be improvable?
>
> $\beta$ is a parameter that controls
> the largest cluster size. It is only needed for the analysis of Algorithm 1 and appears in the upper bound in equation (10) in Theorem 1. We do not expect bound (10) to be optimal and most likely can be improved. Since we only need consistency from Algorithm 1, we have not made an attempt at improving it.
>
> On the other hand, the upper bound in Theorem 2 (equation (12)) should be optimal and it does not depend on $\beta$.  Instead, Theorem 2 assumes that the smallest cluster size ($n\_{\min}$) should not be too small. The dependence on $n\_{\min}$ here is most likely not improvable.
>
> >The kinds of issues the authors run into in terms of cluster recovery are similar to those in the following recent paper ...
>
> Thanks for the pointer. The paper by Gamlath et. al. considers the usual $k$-means clustering based on a set of points $P \subset \mathbb R^d$, but they also also assume that one has side information in the form of a noisy perturbation of the optimal  $k$-means clustering corresponding to $P$. Their goal is to see the impact of this extra information, and they are mainly concerned with obtaining improved constant-factor approximations to the original $k$-means problem. The problem they consider is quite different from the consensus clustering problem we consider here. (They have two sources of information which play asymmetric roles in the problem, one is a set of points, the other a noisy clustering of them, etc. We have $N$ sources of information which play symmetric roles.)
>
> Although it is worth noting that our noisy model for clusterings is the same as what Gamlath et. al. call uniform stochastic noise model.
>
> There might be other similarities in technical details which we cannot comment on without carefully reviewing that paper.

---

### Decision · Program_Chairs · 2023-01-20

**Decision:**

Accept: poster

**Justification For Why Not Higher Score:**

Having seen past oral/spotlight papers, I don't see this rising to that level. For instance, while the dependence on the parameter N is analyzed, the dependence on the other parameters such as beta is not.

**Justification For Why Not Lower Score:**

A clean and natural model, good theoretical progress, some surprises regarding known approximation algorithms.

**Metareview: Summary, Strengths And Weaknesses:**

The paper studies the problem of aggregating clusterings of a dataset: we are given a collection of clusterings of a dataset and the goal is to produce a new clustering that is "close" to the given ones. The problem is non-trivial because clusterings remain the same even after permuting the labels. This work develops a natural algorithm based on studying the association/similarity matrices of the clusterings and taking an average of those. This is a natural idea, and it has been extensively studied in the literature as the authors also note.

The paper is very well-structured and well-written.

The main contributions of the paper are to provide: a consistency analysis of such a procedure under a generative model for clusterings, and an algorithm that can achieve an error that drops optimally with the number of clusterings. The generative model is what the authors call the "robust perturbation model", and is a natural one in this context: a set of clusterings are obtained from a ground-truth clustering by permuting the labels and adding IID error to them. Given these clusterings, the goal is to recover the ground truth up to a permutation.

Given the generative model, the authors show that a very popular algorithm (BestOFK) is in general inconsistent, despite being a 2-approximation to the median partition problem. This shows that consistency in this problem is not to be taken for granted.

Among the questions raised are that since RPM has a strong connection with Bayesian aggregation, have other probabilistic models been considered, and how robust the proposed algorithm is against other settings.



**Note From Pc:**

if the above contains the word "oral" or "spotlight" please see: "oral" presentation means -> notable-top-5% and "spotlight" means -> notable-top-25%. As stated in our emails, we are disassociating presentation type from AC recommendations